# CB$_1$R regulates soluble leptin receptor levels via CHOP, contributing to hepatic leptin resistance

Adi Drori[1], Asaad Gammal[1], Shahar Azar[1], Liad Hinden[1], Rivka Hadar[1], Daniel Wesley[2], Alina Nemirovski[1], Gergő Szanda[3], Maayan Salton[4], Boaz Tirosh[5], Joseph Tam[1]*

[1]Obesity and Metabolism Laboratory, Institute for Drug Research, School of Pharmacy, Faculty of Medicine, The Hebrew University of Jerusalem, Jerusalem, Israel; [2]Laboratory of Physiological Studies, National Institute on Alcohol Abuse & Alcoholism, Bethesda, United States; [3]MTA-SE Laboratory of Molecular Physiology, Department of Physiology, Semmelweis University, Budapest, Hungary; [4]Department of Biochemistry and Molecular Biology, The Institute for Medical Research Israel-Canada, The Hebrew University of Jerusalem, Jerusalem, Israel; [5]The Institute for Drug Research, School of Pharmacy, Faculty of Medicine, The Hebrew University of Jerusalem, Jerusalem, Israel

**Abstract** The soluble isoform of leptin receptor (sOb-R), secreted by the liver, regulates leptin bioavailability and bioactivity. Its reduced levels in diet-induced obesity (DIO) contribute to hyperleptinemia and leptin resistance, effects that are regulated by the endocannabinoid (eCB)/CB$_1$R system. Here we show that pharmacological activation/blockade and genetic overexpression/deletion of hepatic CB$_1$R modulates sOb-R levels and hepatic leptin resistance. Interestingly, peripheral CB$_1$R blockade failed to reverse DIO-induced reduction of sOb-R levels, increased fat mass and dyslipidemia, and hepatic steatosis in mice lacking C/EBP homologous protein (CHOP), whereas direct activation of CB$_1$R in wild-type hepatocytes reduced sOb-R levels in a CHOP-dependent manner. Moreover, CHOP stimulation increased sOb-R expression and release via a direct regulation of its promoter, while CHOP deletion reduced leptin sensitivity. Our findings highlight a novel molecular aspect by which the hepatic eCB/CB$_1$R system is involved in the development of hepatic leptin resistance and in the regulation of sOb-R levels via CHOP.

*For correspondence:
yossi.tam@mail.huji.ac.il

Competing interests: The authors declare that no competing interests exist.

## Introduction

Leptin, predominantly produced by and secreted from white adipocytes, conveys information regarding the status of energy storage and availability to the brain to maintain energy homeostasis. It binds the leptin receptor in hypothalamic neurons to reduce food intake and increase energy expenditure in coordination with other adipokines and gastric peptides (*Allison and Myers, 2014*; *Pan and Myers, 2018*). Molecularly, leptin stimulates the secretion of α-melanocortin stimulating hormone (α-MSH) from proopiomelanocortin (POMC) neurons at the arcuate nucleus (ARC) and inhibits the secretion of the orexigenic peptides neuropeptide-Y (NPY) and Agouti-related protein (AgRP) (*Flak and Myers, 2016*). Genetic leptin deficiency or lack of functional leptin receptor results in morbid obese and insulin resistance phenotypes in animals (*Lep$^{ob/ob}$* or *Lepr$^{db/db}$* mice, respectively) (*Tartaglia et al., 1995*; *Zhang et al., 1994*). In humans, congenital leptin deficiencies are rare, leading to hyperphagia and early-onset obesity, which can be reversed with a leptin replacement

**eLife digest** When the human body has stored enough energy from food, it releases a hormone called leptin that travels to the brain and stops feelings of hunger. This hormone moves through the bloodstream and can affect other organs, such as the liver, which also help control our body's energy levels. Most people with obesity have very high levels of leptin in their blood, but are resistant to its effects and will therefore continue to feel hungry despite having stored enough energy.

One of the proteins that controls the levels of leptin is a receptor called sOb-R, which is released by the liver and binds to leptin as it travels in the blood. Individuals with high levels of this receptor often have less free leptin in their bloodstream and a lower body weight. Another protein that helps the body to regulate its energy levels is the cannabinoid-1 receptor, or CB1R for short. In people with obesity, this receptor is overactive and has been shown to contribute to leptin resistance, which is when the brain becomes less receptive to leptin. Previous work in mice showed that blocking CB1R reduced the levels of leptin and allowed mice to react to this hormone normally again, but it remained unclear whether CB1R affects how other organs, such as the liver, respond to leptin.

To answer this question, Drori et al. blocked the CB1R receptor in the liver of mice eating a high-fat diet, either by using a drug or by deleting the gene that codes for this protein. This caused mice to have higher levels of sOb-R circulating in their bloodstream. Further experiments showed that this change in sOb-R was caused by the levels of a protein called CHOP increasing in the liver when CB1R was blocked. Drori et al. found that inhibiting CB1R caused these obese mice to lose weight and have healthier, less fatty livers as a result of their livers no longer being resistant to the effects of leptin.

Scientists, doctors and pharmaceutical companies are trying to develop new strategies to combat obesity. The results from these experiments suggest that blocking CB1R in the liver could allow this organ to react to leptin appropriately again. Drugs blocking CB1R, including the one used in this study, will be tested in clinical trials and could provide a new approach for treating obesity.

therapy (*Mantzoros, 1999*). However, most cases of obesity are characterized by hyperleptinemia, indicating that obesity is a leptin-resistant state, where leptin signaling is impaired.

Whereas many of the actions of leptin are attributed to its effects in the brain, it also has a broad range of physiological effects in the periphery such as angiogenesis, bone formation, lipid and carbohydrate metabolism, nutrient absorption, and insulin homeostasis (*Sáinz et al., 2015*). In fact, the lack of a response to leptin due to the development of resistance to the hormone may directly affect the central and peripheral actions of leptin, leading to a dysregulated energy balance. For instance, the liver, a central organ in the regulation of whole-body energy homeostasis, constitutes an important target for leptin as it regulates hepatic gluconeogenesis and insulin sensitivity as well as lipid metabolism (*Frühbeck and Nutr, 2002*). Therefore, defects in leptin action, which occur in a state of hepatic leptin resistance, impair hepatic function and lead to hyperglycemia, hyperinsulinemia, and dyslipidemia (*Frühbeck and Nutr, 2002*).

Various mechanisms have been linked to the development of diet-induced obesity (DIO)-related central and peripheral leptin resistance, including limited CNS access of leptin due to saturated transport machinery, uncoupling of leptin from its receptor (due to rare genetic mutations or intracellular modulators), leptin-induced downregulation of its hypothalamic receptor, and several circulating factors such as the soluble isoform of leptin receptor (sOb-R) (reviewed in *Engin, 2017*; *Martin et al., 2008*). Both in humans and mice, the leptin receptor gene (*LEPR* and *Lepr*, respectively) encodes four membrane-anchored isoforms, which differ in the length of their cytoplasmic tail. The long isoform, Ob-Rb, is considered to convey the most robust cellular response to leptin, while the shorter isoforms (Ob-Ra, Ob-Rc, and Ob-Rd) carry a weaker signal. In addition, sOb-R, which lacks the trans-membrane and intracellular domains, also exists. In humans, sOb-R is exclusively generated via proteolytic shedding of membrane-anchored isoforms (*Maamra et al., 2001*), whereas in mice, it is produced by both transcription of a designated isoform (Ob-Re) and ectodomain shedding of Ob-Rb and Ob-Ra (*Ge et al., 2002*; *Li et al., 1998*). sOb-R, mainly produced by hepatocytes, is the main leptin-binding protein in human plasma, regulating leptin's bioavailability

and bioactivity (*Lammert et al., 2001*; *Yang et al., 2004*). In fact, studies have shown that the circulating levels of sOb-R are inversely correlated with body weight and free leptin levels (*Ogier et al., 2002*). In addition, sOb-R levels are increased following weight loss (*Laimer et al., 2002*; *Reinehr et al., 2005*), and its overexpression in mice increases leptin sensitivity (*Huang et al., 2001*; *Lou et al., 2010*), supporting the key role of sOb-R in the development as well as the reversal of leptin resistance.

The endocannabinoid (eCB) system, a major regulator of energy homeostasis (*Cristino et al., 2014*; *Fride et al., 2005*; *Pagotto et al., 2006*; *Ruiz de Azua and Lutz, 2019*; *Silvestri and Di Marzo, 2013*), evokes various cellular/metabolic pathways via the activation of two G-protein-coupled receptors, cannabinoid type-1 ($CB_1R$) and type-2 ($CB_2R$) receptors, by the main eCBs, *N*-arachidonoylethanolamine (AEA) and 2-arachidonoylglycerol (2-AG). The eCB/$CB_1R$ system is highly overactive during obesity (*Engeli, 2008*; *Engeli et al., 2005*; *Matias et al., 2006*; *Monteleone et al., 2005*), and both central and peripheral stimulations of this system have been suggested to contribute to the development of the metabolic syndrome, including leptin resistance (*Engeli et al., 2005*; *Matias et al., 2008*; *Pagotto et al., 2005*). Studies have shown that leptin's ability to regulate food intake and peripheral lipid metabolism depends upon hypothalamic $CB_1Rs$ (*Buettner et al., 2008*; *Cardinal et al., 2014*; *Di Marzo et al., 2001*; *Jo et al., 2005*; *Malcher-Lopes et al., 2006*). Recent evidence demonstrates that peripheral $CB_1R$ signaling has the ability to modulate leptin activity too. By using peripherally restricted $CB_1R$ blockers, we have recently demonstrated that DIO-related hyperleptinemia is completely reversed by increasing leptin's renal clearance and decreasing its secretion from adipocytes (*Tam et al., 2012*; *Tam et al., 2010*). Additionally, we have shown that the reversal of hypothalamic leptin resistance in obese mice treated with the peripherally restricted $CB_1R$ blocker, JD5037, is mediated via re-sensitizing the animals to endogenous leptin and re-activating POMC neurons (*Tam et al., 2017*). Several lines of evidence suggest that hypothalamic neurons, including POMC, undergo endoplasmic reticulum (ER) stress during DIO, which may contribute to the development of leptin resistance (*Ozcan et al., 2009*; *Ramírez and Claret, 2015*). We have previously reported that pharmacological inhibition of peripheral $CB_1Rs$ (by AM6545) reverses the high-fat diet (HFD)-induced hepatic elevation in the ER stress marker phospho-eIF2α (*Tam et al., 2010*). Since ER stress strongly affects protein translation and secretion (reviewed in *Ron and Walter, 2007*), we hypothesized that the eCB/$CB_1R$ system plays a direct role in the regulation of sOb-R levels and hepatic leptin signaling involves the ER stress signaling pathway.

## Results

### Hepatic $CB_1R$ regulates sOb-R levels and leptin signaling

To evaluate the direct contribution of $CB_1R$ to the regulation of sOb-R levels, we first utilized a pharmacological inhibition paradigm of $CB_1R$ in DIO mice by using the peripherally restricted $CB_1R$ inverse agonist JD5037. Similar to previous findings (*Mazor et al., 2018*), a significant reduction in serum levels of sOb-R was documented in obese mice, an effect that was ameliorated by JD5037 treatment (*Figure 1A*). Since sOb-R is mainly produced by the liver (*Lammert et al., 2001*), we also analyzed the content of sOb-R in liver homogenates from these animals and found a similar trend as in serum (*Figure 1B*). Measurements of the *Lepr-s* (*Ob-Re)* isoform revealed that JD5037 treatment also affected its transcription and protein levels (*Figure 1C–E*). Moreover, the protein expression of two additional isoforms of LEPR (Ob-Rb and Ob-Ra) in liver homogenates was also decreased in DIO mice and normalized following JD5037 treatment (*Figure 1D–E*).

To further establish the contribution of hepatic $CB_1R$ to the HFD-induced decrease in sOb-R levels, we utilized the liver-specific $CB_1R$ null (LCB1 cKO) mice, a genetic deletion model of mice that lacks $CB_1R$ specifically in hepatocytes (mouse model generation is described in *Osei-Hyiaman et al., 2008*). When fed with a HFD, these mice gain similar weight to their wild-type (WT) littermate controls [(*Osei-Hyiaman et al., 2008*) and *Figure 1F*]; however, they are less prone to develop liver steatosis, dyslipidemia, and leptin resistance (*Osei-Hyiaman et al., 2008*), making hepatic $CB_1R$ a central regulator of obesity-related liver complications. We were therefore not surprised to find that the liver specific deletion of $CB_1R$ was sufficient to maintain normal circulating levels of sOb-R in obese LCB1 cKO mice (*Figure 1A*). Similarly, the hepatic gene and protein expression of sOb-R and

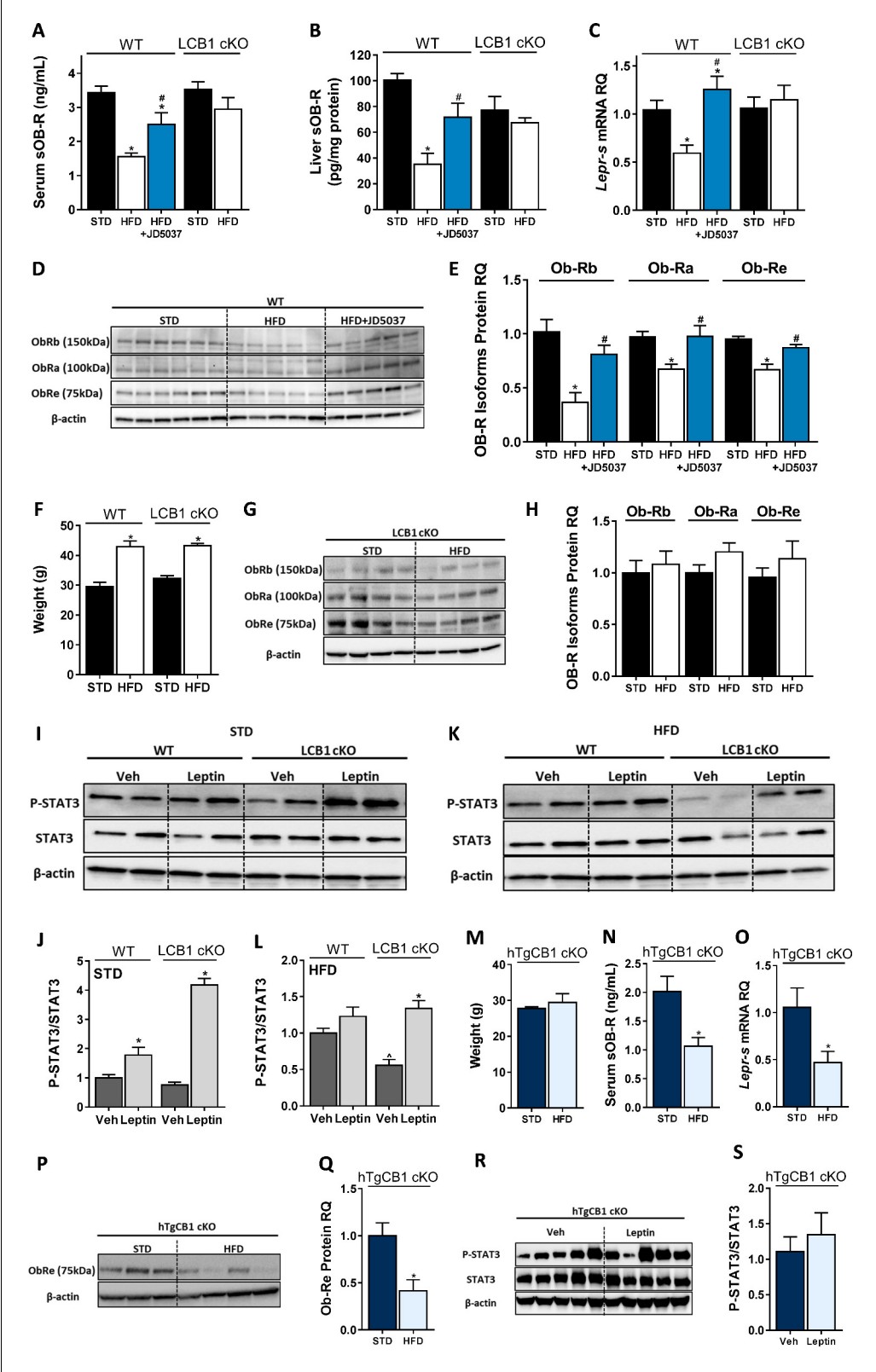

**Figure 1.** Hepatic $CB_1R$ regulates soluble isoform of leptin receptor (sOb-R) levels and leptin signaling in DIO. Serum (**A**, n = 7–21) and liver (**B**, n = 6–9) levels of sOb-R are reduced following a 14 weeks consumption of high-fat diet (HFD) in wild-type (WT), but not LCB1 cKO mice. JD5037 (3 mg/kg, for 7 days) reverses the reduction in WT mice. The same trend observed in hepatic mRNA levels of *Lepr-s* (**C**, n = 3–12) as well as protein levels of Ob-Rb, Ob-Ra, and Ob-Re (**D and E**, n = 5–10; **G and H**, n = 8–10). Despite a comparable weight gain following HFD consumption in WT and LCB1 cKO mice

*Figure 1 continued on next page*

*Figure 1 continued*

(**F**, n = 4–20), obese mice that lack $CB_1R$ in hepatocytes remain leptin sensitive indicated by increased pSTAT3 levels (**I–L**, n = 3–4). Transgenic mice, expressing $CB_1R$ only in hepatocytes, are protected from DIO (**M**, n = 3–4). An exclusive hepatic overexpression of $CB_1R$ is sufficient for HFD feeding to induce reduction in serum, liver mRNA, and protein levels of sOb-R (**N–Q**, n = 4) as well as to reduce hepatic leptin sensitivity in response to exogenous leptin stimulation (**R and S**, n = 7). Data represent mean ± SEM of indicated number of replicates in each panel. The blots are representative. *$p<0.05$ relative to standard diet-fed animals from the same strain. #$p<0.05$ relative to HFD-fed mice from the same strain. ˆ$p<0.05$ relative to the same treatment group of WT mice.

The online version of this article includes the following source data and figure supplement(s) for figure 1:

**Source data 1.** Raw data for *Figure 1*.
**Figure supplement 1.** LEPR antibody detects isoforms of 75, 100, and 150 kDa.
**Figure supplement 1—source data 1.** Raw data for *Figure 1—figure supplement 1*.

the other LEPR isoforms were not affected by the HFD feeding (*Figure 1B,C and G,H*), suggesting that hepatic $CB_1R$ most likely regulates sOb-R levels.

To test the functional relevance of our findings to hepatic leptin signaling, we measured the phosphorylation levels of STAT3, the gold-standard measure of leptin signaling (reviewed in *Allison and Myers, 2014*), in mouse livers following exogenous leptin administration in vivo. Whereas both lean WT and LCB1 cKO mice showed elevated pSTAT3/STAT3 ratio in response to leptin (*Figure 1I,J*), only obese LCB1 cKO mice remained leptin sensitive (*Figure 1K,L*). These results are in line with findings from Osei-Hyiaman and colleagues (*Osei-Hyiaman et al., 2008*), demonstrating that LCB1 cKO mice are resistant to obesity-induced hyperleptinemia.

Additional support for the regulation of sOb-R by hepatic $CB_1R$ derived from another transgenic mouse model (hTgCB1 cKO), in which $CB_1R$ is expressed only in hepatocytes (mouse model generation is described in *Liu et al., 2012*; *Tam et al., 2010*). These mice, despite being resistance to DIO like global $CB_1R$ KO mice [(*Liu et al., 2012*; *Tam et al., 2010*) and *Figure 1M*], demonstrate increased circulating leptin levels when fed a HFD (*Liu et al., 2012*). In accordance with that, the circulating and hepatic sOb-R levels in these mice were decreased by 50% following 14 weeks consumption of a HFD (*Figure 1N–Q*). Moreover, the hepatic pSTAT3/STAT3 ratio did not respond to exogenous leptin, suggesting reduced hepatic leptin sensitivity (*Figure 1R,S*). Hence, overexpression of $CB_1R$ in the liver alone compromises hepatic leptin sensitivity and recapitulates the HFD-induced downregulation of sOb-R observed in WT mice.

Next, we assessed whether a direct activation of $CB_1R$ in hepatocytes induces a reduction in sOb-R levels. To test this, we treated cultured hepatocytes with the synthetic $CB_1R$ agonist noladin ether (NE) for 24 hr. We analyzed both culture media and cell lysates and found that, similar to obesity, direct activation of $CB_1R$ also decreased sOb-R levels in the culture media. This was also the case with intracellular levels of other LEPR isoforms measured. This $CB_1R$-mediated reduction in ObR levels was completely reversed by blocking $CB_1R$ using JD5037 (*Figure 2*).

## CHOP contributes to the metabolic response to peripheral $CB_1R$ blockade

DIO-induced ER stress in the development of leptin resistance has been previously suggested (*Ozcan et al., 2009*; *Ramírez and Claret, 2015*). Similar to our previous findings (*Tam et al., 2010*), treatment of HFD-fed mice with JD5037 normalized p-eIF2α levels (*Figure 3—figure supplement 1A,B*), suggesting relieved ER stress following $CB_1R$ blockade. In agreement with these findings, a comparable ratio of hepatic phospho-to-total eIF2α ratio was documented in lean and obese LCB1 cKO mice (*Figure 3—figure supplement 1C,D*).

Measuring the expression levels of the ER stress marker C/EBP homologous protein (CHOP) revealed surprising findings, since both the hepatic mRNA (*Ddit3*) and protein levels of CHOP were downregulated in obese WT mice, despite the suggested ER stress. Its expression levels were reversed above control levels by JD5037, and remained comparable between lean and obese LCB1 cKO mice (*Figure 3A–D*). In fact, CHOP levels were positively correlated with the levels of sOb-R in both our experimental paradigms, leading us to hypothesize that CHOP may directly be involved in the regulation of sOb-R.

To test our hypothesis, we compared the metabolic efficacy of JD5037 in obese CHOP KO mice and their littermate controls. Whereas JD5037 was almost equieffective in reducing body weight

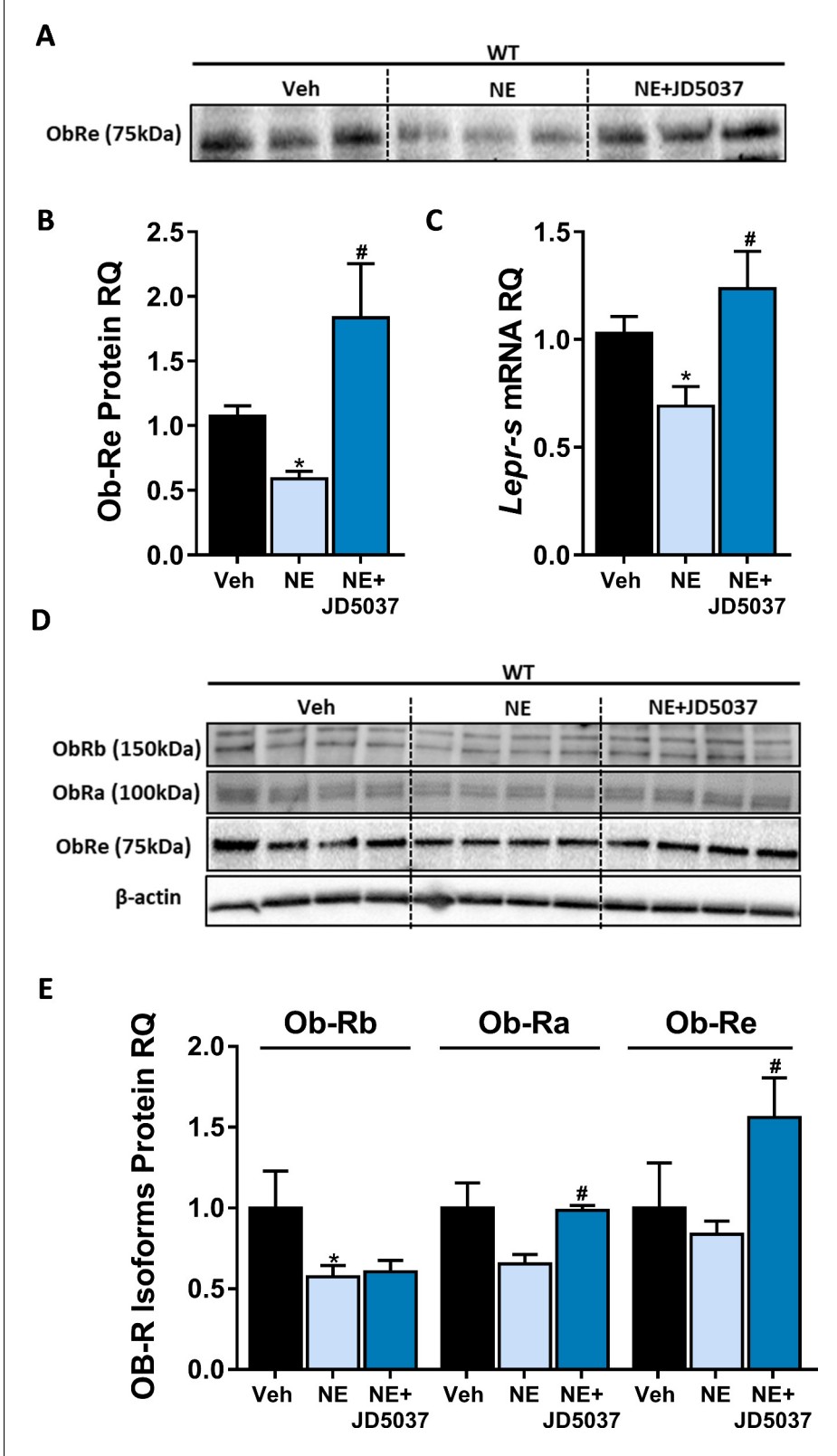

**Figure 2.** CB$_1$R directly regulates soluble isoform of leptin receptor (sOb-R) levels in hepatocytes. 24 hr treatment with the synthetic CB$_1$R agonist noladin ether (NE, 2.5 µM) induced reduction in sOb-R levels in the culture media of immortalized hepatocytes (blot was quantified using Ponceau staining as a loading control). This was completely ameliorated by 1 hr pretreatment with 100 nM JD5037 (**A and B**, n = 12–13). Similar results were observed in both mRNA (**C**, n = 12–17) and protein (**D and E**, n = 4) levels in hepatocytes lysate (for Ob-Rb, the lower band was quantified). Data

*Figure 2 continued on next page*

*Figure 2 continued*

represent mean ± SEM of indicated number of biological replicates. Blots are representative. *p<0.05 relative to vehicle-treated cells. #p<0.05 relative to NE-treated cells.

The online version of this article includes the following source data and figure supplement(s) for figure 2:

**Source data 1.** Raw data for *Figure 2*.
**Figure supplement 1.** Hepatic expression of ADAM10 and ADAM17.
**Figure supplement 1—source data 1.** Raw data for *Figure 2—figure supplement 1*.

and fat mass in both obese mouse strains (*Figure 3E–G*), it improved plasma cholesterol levels as well as hepatic steatosis in WT mice only (*Figure 3H–J*). The reduced ability of peripherally restricted CB$_1$R blockade to improve dyslipidemia and hepatic steatosis in CHOP KO mice led us to measure the hepatic eCB 'tone' in these mice. Strikingly, we found that the basal levels of AEA and 2-AG were markedly higher in CHOP KO mice than in the WT control group. Moreover, the increased eCB levels in CHOP KO mice remained unchanged following a consumption of HFD as well as JD5037 treatment (*Figure 3—figure supplement 2A,B*). This could be partially explained by the differences documented in the mRNA expression patterns of fatty acid amide hydrolase (*Faah*), monoacylgly-cerol lipase (*Mgll*), N-acyl phosphatidylethanolamine phospholipase D (*Napepld*), and diacylglycerol lipase alpha (*Dagla*), the degrading and synthesizing enzymes of both eCBs, respectively (*Figure 3—figure supplement 2C–F*). Overall, these data indicate that CHOP may play a pivotal role in modu-lating hepatic eCB 'tone', and that it is required for the beneficial effects of CB$_1$R blockade on dysli-pidemia and hepatic steatosis.

## CHOP plays a key role in the regulation of sOb-R by the eCB/CB$_1$R system

Measuring the effect of CHOP deficiency on sOb-R levels revealed comparable circulating levels of sOb-R in lean and obese mice in the two mouse strains. However, JD5037 failed to restore sOb-R levels in CHOP KO mice (*Figure 4A*). The assessment of *Lepr-s* mRNA expression and sOb-R protein content in the livers of both strains documented reduced baseline levels in CHOP KO mice, com-pared to WT, which still remained low following HFD consumption and/or JD5037 treatment (*Figure 4B,C*). A similar trend was observed in the protein level of two more LEPR isoforms (Compare *Figure 1D, E* to *Figure 4D,E*). The HFD-induced hyperleptinemia was vastly reduced by JD5037 treatment in WT mice, whereas it was only partially ameliorated by JD5037 in CHOP KO ani-mals (*Figure 4F*). Interestingly, the hepatic pSTAT3/STAT3 ratio in lean CHOP KO mice was compa-rable before and after stimulation with exogenous leptin (*Figure 4G,H*). Taken together, our data suggest that the regulation of sOb-R levels is CHOP-dependent. In addition, regulation of the solu-ble isoform by CHOP can consequently affect circulating leptin levels and hepatic leptin sensitivity, possibly, in a CB$_1$R-dependent manner.

To further investigate this concept, we directly activated CB$_1$R (with NE) in immortalized hepato-cytes originated from WT or CHOP KO mice. Similar to a HFD consumption in mice (*Figure 3A*), a direct activation of CB$_1$R downregulated CHOP mRNA expression (*Figure 5A*). We validated this by measuring the expression levels of *Ppp1r15a,* a downstream target of CHOP (*Hu et al., 2018*), and found that its expression was also reduced in NE-treated WT hepatocytes, and remained unchanged in CHOP KO cells (*Figure 5B*), suggesting that CB$_1$R activation in fact leads to reduced CHOP expression and activity. Whereas NE was able to reduce sOb-R levels in WT hepatocytes, it had the opposite effect in CHOP KO hepatocytes, suggesting that CB$_1$R may regulate sOb-R levels in other mechanisms independently of CHOP (*Figure 5C–E*).

The consistent correlation between CHOP and sOb-R levels implies that CHOP is a positive regu-lator of Ob-Re. To validate this further, we analyzed Ob-Re levels in WT and CHOP KO hepatocytes treated with tunicamycin (TM), a potent inducer of ER stress. Treatment with TM for 6 hr led to an expected and robust expression of CHOP mRNA and protein in WT cells (*Figure 6A,B*). Importantly, this was accompanied with elevated mRNA expression levels of *Lepr-s* as well as secreted levels of sOb-R into the culture media in WT, but not CHOP KO hepatocytes (*Figure 6C–E*). Increased levels of sOb-R in culture media were also documented when we exogenously overexpressed myc-tagged CHOP in WT hepatocytes (*Figure 6F,G*), supporting a direct role for CHOP in *Lepr* gene regulation.

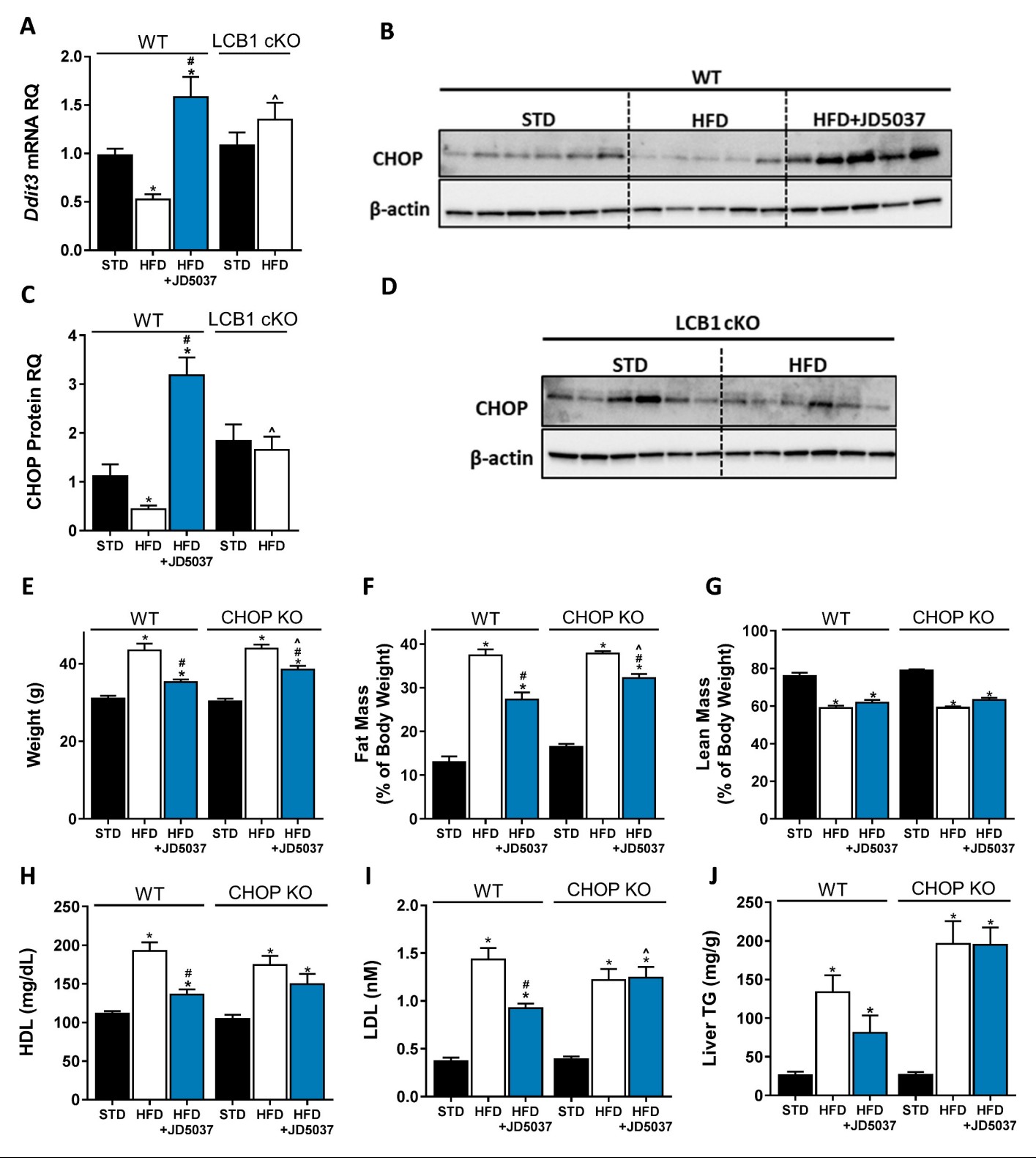

**Figure 3.** C/EBP homologous protein (CHOP) contributes to the metabolic benefits of peripheral CB$_1$R blockade. mRNA (**A**, n = 5–19) and protein (**B–D**, n = 4–6) levels of CHOP show reduced expression following 14 weeks on high-fat diet (HFD) in wild-type (WT), but not LCB1 cKO mice. JD5037 (3 mg/kg, for 7 days) treatment reverses the HFD-induced reduction in CHOP levels. Metabolic assessment of mice revealed diminished effect of JD5037 in CHOP KO mice. Weight (**E**, n = 10–23), fat and lean mass (**F and G**, n = 5–19), serum HDL, LDL as well as hepatic triglycerides (TG) (**H–J**, n = 9–26) were comparable in lean and obese WT and CHOP KO mice. JD5037 treatment was significantly more efficient in reducing weight, fat mass, LDL, and

*Figure 3 continued on next page*

*Figure 3 continued*

TG in WT mice. Data represent mean ± SEM of indicated number of replicates in each panel. *p<0.05 relative to standard diet-fed animals from the same strain. #p<0.05 relative to HFD-fed mice from the same strain. ̂p<0.05 relative to the same treatment group of WT mice.

The online version of this article includes the following source data and figure supplement(s) for figure 3:

**Source data 1.** Raw data for *Figure 3*.
**Figure supplement 1.** High-fat diet (HFD) induces eIF2α phosphorylation in wild-type (WT) mice.
**Figure supplement 1—source data 1.** Raw data for *Figure 3—figure supplement 1*.
**Figure supplement 2.** An altered endocannabinoid (eCB) 'tone' in C/EBP homologous protein (CHOP)-deficient mice.
**Figure supplement 2—source data 1.** Raw data for *Figure 3—figure supplement 2*.

By using a luciferase reporter assay, in which the −650 to +850 (relative to transcription start site) region of the LEPR promoter was cloned into firefly luciferase expressing vector, we found that CHOP expression and luciferase activity in transfected cells was induced using TM (*Figure 6H*), while CB$_1$R activation using NE (which downregulates CHOP expression as seen in *Figure 5A*) had an opposite effect in WT, but not in CHOP KO cells. These data support the involvement of CHOP in CB$_1$R-dependent regulation of sOb-R.

To further explore the possibility that CHOP can directly bind the *Lepr* promoter and control its expression, we performed several chromatin immunoprecipitation (ChIP) experiments. In silico analysis of the *Lepr* promoter region revealed a putative binding site, corresponding to five of six nucleotides that compose a core sequence for CHOP binding (GRCm38:CM000997.2. Chromosome 4: 101,717,929–101,717,934) (*Ubeda et al., 1996*). As seen in the CHOP precipitates (*Figure 6I*), there was a twofold increase in the recovery of the qPCR product amplified with a primer set flanking the putative CHOP binding site, in cells that were treated with TM. A similar enrichment was seen in *Ppp1r15a* (GADD34), a well-known target of CHOP. This increase was limited to WT hepatocytes, validating the specificity of CHOP IP. Taken together, our data suggests that CHOP is able to occupy the *Lepr* promoter and directly regulate sOb-R levels in response to HFD consumption and/or CB$_1$R activation.

The molecular signaling pathway(s) by which eCBs/CB$_1$R regulates CHOP levels calls for further investigation. Nevertheless, one putative mechanism may involve Trib3, a multifunctional protein upregulated during ER stress by the PERK-ATF4-CHOP pathway, which mediates cell death. Trib3 represses its own expression by inhibiting the transcription of both ATF4 and CHOP (*Jousse et al., 2007*; *Mathur et al., 2014*; *Ohoka et al., 2005*). In addition, many studies describe Trib3 as a key factor in mediating the anti-tumor effect of cannabinoids (reviewed in *Velasco et al., 2016*). Our in vivo data indicate that HFD induces the mRNA and protein expression levels of hepatic Trib3, and that treatment with JD5037 restores these levels. This effect is limited to WT mice, whereas in CHOP KO mice, Trib3 levels did not change in response to HFD nor JD5037 treatment. Similarly, a direct activation of CB$_1$R using NE upregulated Trib3 expression in WT, but not in CHOP KO hepatocytes (*Figure 6—figure supplement 1A–E*), suggesting that Trib3 is indeed induced via CB$_1$R signaling, and negatively regulates CHOP levels. Further support for this hypothesis comes from our data in *Figure 6—figure supplement 2*, where we show that ATF4 protein levels are reduced in WT mice following the consumption of HFD, and are normalized by JD5037, whereas they remain unchanged in lean and obese LCB1 cKO mice (*Figure 6—figure supplement 2B–D*). Moreover, the ATF4 levels were reduced in hTgCB1 cKO mice fed with a HFD, as compared to lean STD-fed mice (*Figure 6—figure supplement 2E–G*). Overall, the CB$_1$R-related changes in ATF4 expression were found to be well correlated with CHOP as well as with the sOb-R levels, placing ATF4 downstream of CB$_1$R and upstream of CHOP in this cascade.

## Discussion

Since only free leptin crosses the blood–brain barrier (BBB) and induces leptin signaling, the sOb-R, which sequesters free leptin in the serum and is considered as the main binding protein for leptin in the circulation, practically regulates leptin's bioavailability and activity and can potentially affect leptin sensitivity/resistance. This is also true for peripheral tissues, where sOb-R/leptin complexes cannot bind to and activate membrane anchored leptin receptors. Many human and animal studies have demonstrated that sOb-R levels are inversely correlated with plasma levels of leptin, BMI, and

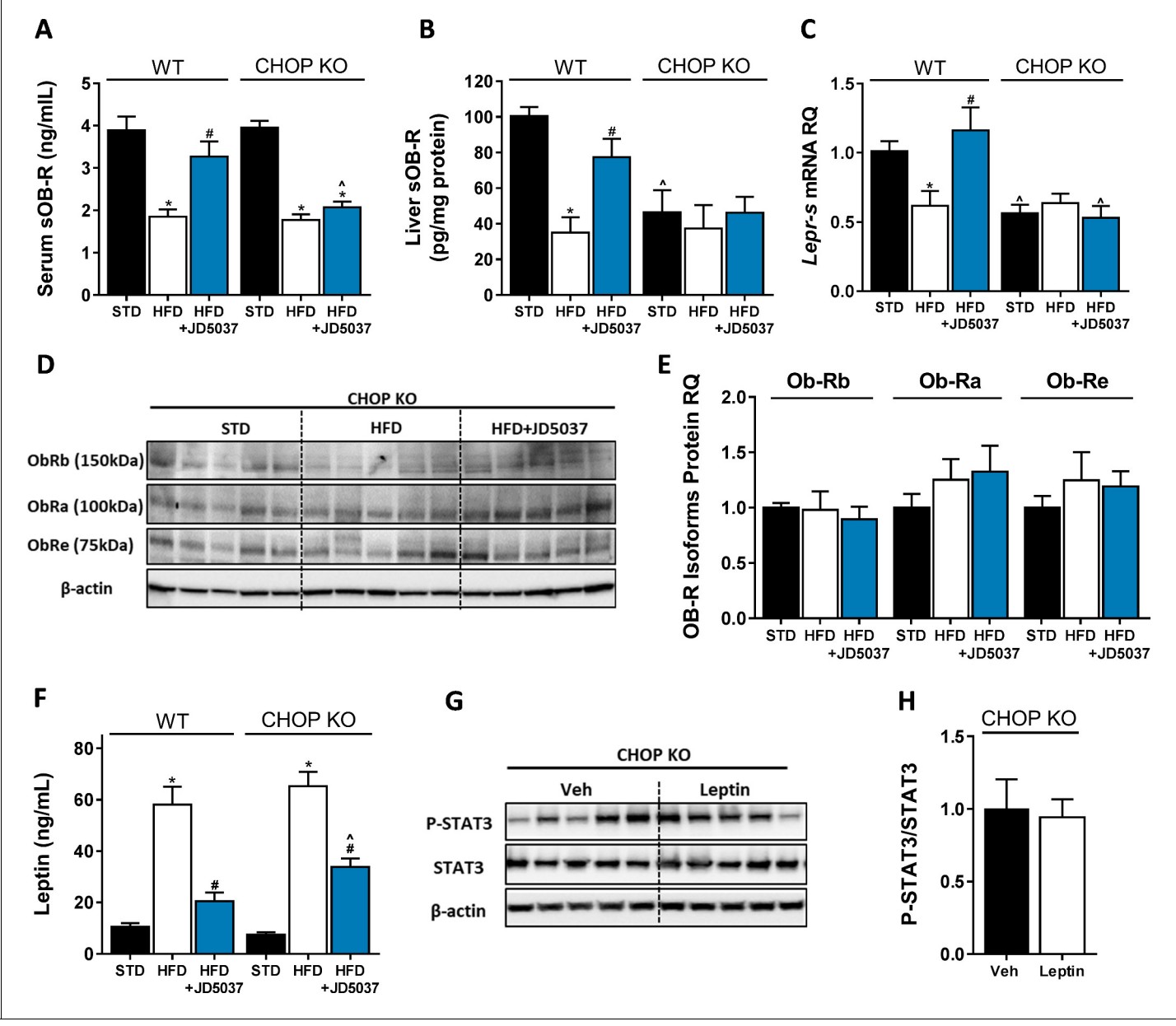

**Figure 4.** C/EBP homologous protein (CHOP) plays a key role in regulating soluble isoform of leptin receptor (sOb-R) by the endocannabinoid (eCB)/ CB$_1$R system. Serum (A, n = 11–18) levels of sOb-R were reduced following a 14 weeks consumption of high-fat diet (HFD). JD5037 (3 mg/kg, for 7 days) reversed the reduction in wild-type (WT), but not CHOP KO mice. Basal hepatic levels of sOb-R were lower in CHOP KO mice and did not change following HFD or JD5037 treatment (B, n = 5–8). A similar trend was observed in hepatic mRNA levels (C, n = 9–17) and protein level of Ob-Rb, Ob-Ra, and Ob-Re (D and E, n = 5–6). Whereas DIO-related hyperleptinemia was comparable between WT and CHOP KO mice, JD5037 was more efficacious in reducing it in WT mice (F, n = 8–16). Lean CHOP KO mice failed to increase the hepatic pSTAT3/STAT3 ratio in response to exogenous leptin administration (G and H, n = 5). Western blots are representative. Data represent mean ± SEM of indicated number of replicates in each panel. *p<0.05 relative to standard diet-fed animals from the same strain. #p<0.05 relative to HFD-fed mice from the same strain. ˆp<0.05 relative to the same treatment group of WT mice.

The online version of this article includes the following source data for figure 4:

**Source data 1.** Raw data for *Figure 4*.

adiposity (*Chan et al., 2002*; *Lahlou et al., 2000*; *Laimer et al., 2002*; *Ogier et al., 2002*; *Reinehr et al., 2005*), suggesting that low levels of the soluble isoform contribute to obesity-related hyperleptinemia and subsequently, leptin resistance. In contrast to pathological conditions with a

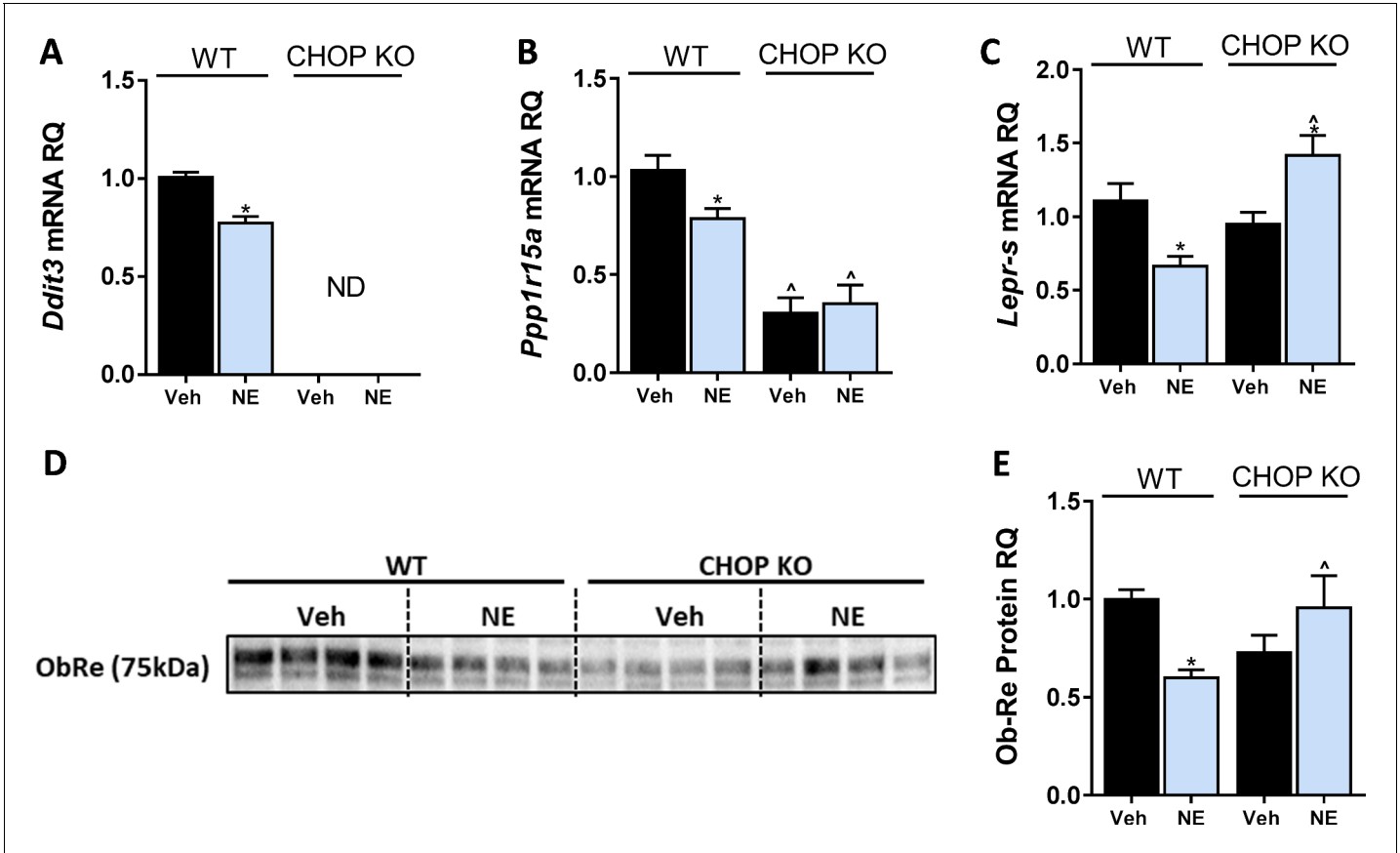

**Figure 5.** C/EBP homologous protein (CHOP) regulates LEPR expression in hepatocytes. In vitro, 24 hr treatment with noladin ether (NE; 2.5 μM) induced reduction in mRNA levels of *Ddit3.* (ND – not detected) (**A**), *Ppp1r15a* (**B**), and *Lepr-s* (**C**) in wild-type (WT), but not CHOP KO hepatocytes. A similar trend was observed in soluble isoform of leptin receptor protein levels secreted into the culture media of hepatocytes (blot was quantified using Ponceau staining as a loading control) (**D and E**). Data represent mean ± SEM of 8–20 biological replicates. Blots are representative. *p<0.05 relative to vehicle-treated cells in the same genotype. ̂p<0.05 relative to same treatment paradigm in WT.

The online version of this article includes the following source data for figure 5:

**Source data 1.** Raw data for *Figure 5*.

positive energy balance (i.e. obesity), human clinical situations associated with energy deficiency (i.e. starvation and/or anorexia nervosa) are characterized by upregulated circulating levels of sOb-R (*Monteleone et al., 2002*; *Reinehr et al., 2005*; *Stein et al., 2006*; *Zepf et al., 2012*). Moreover, individuals carrying a mutated allele of *LEPR*, which leads to enhanced shedding of the leptin binding domain, have normoleptinemia and they are not obese (*Lahlou et al., 2002*). For these reasons, the sOb-R most likely plays a key role in the formation of central and peripheral leptin resistance conditions. Yet, only limited knowledge exists about the molecular mechanisms that regulate sOb-R production and secretion.

Using multiple cultured cell types, Gan and colleagues have shown that TNFα may induce cell surface expression of Ob-Rb as well as sOb-R levels (*Gan et al., 2012*). In addition, an in vitro study has demonstrated that increasing the concentration of recombinant sOb-R diminishes STAT3 phosphorylation in response to leptin stimulation, but pre-incubation of leptin with recombinant sOb-R forms ligand-receptor complexes do not affect leptin-mediated STAT3 phosphorylation (*Yang et al., 2004*). In vivo, it has been described that leptin stimulation as well as food deprivation specifically induce the expression of sOb-R in mouse liver (*Cohen et al., 2005*). It has been also demonstrated that in contrast to mice, the human sOb-R, exclusively generated through proteolytic cleavage of the extracellular domain of membrane-anchored isoforms (*Maamra et al., 2001*), is shed into the circulation by two well-known proteolytic enzymes, ADAM10 and ADAM 17, belong to the 'ADAM's family' (reviewed in *Schaab and Kratzsch, 2015*). As we could not detect significant up/down

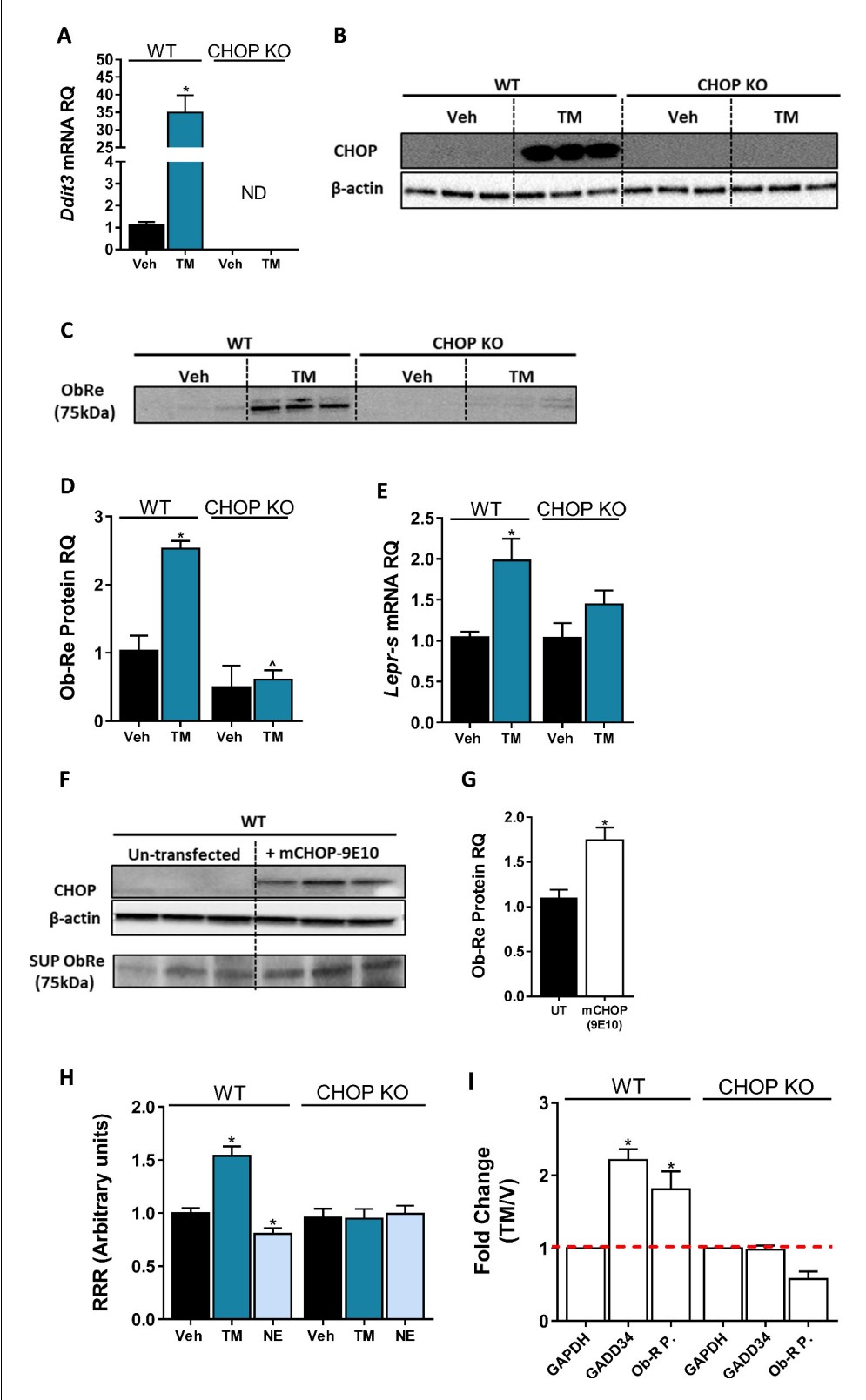

**Figure 6.** C/EBP homologous protein (CHOP) is a positive regulator of *Lepr* Promoter. Induction of CHOP mRNA (**A**, n = 14) and protein (**B**, n = 3) expression using a 6 hr treatment with tunicamycin (TM; 2.5 μG/mL) was accompanied by elevated soluble isoform of leptin receptor (sOb-R) levels in the culture media (blot was quantified using Ponceau staining as a loading control) (**C and D**, n = 3) as well as mRNA levels (**E**, n = 8–21) of wild-type (WT) hepatocytes. A transient CHOP overexpression induced elevation in sOb-R levels in the culture media of WT hepatocytes (**F and G**, n = 3). *Figure 6 continued on next page*

*Figure 6 continued*

Luciferase reporter assay (H, n = 14–16) and chromatin immunoprecipitation (ChIP) (I, n = 2–7) show increased *Lepr* promoter activity and CHOP binding to this promoter in WT hepatocytes treated with TM. Data represent mean ± SEM of indicated number of biological replicates. Blots are representative. *p<0.05 relative to vehicle-treated cells. ʰp<0.05 relative to the same treatment group of WT hepatocytes.

The online version of this article includes the following source data and figure supplement(s) for figure 6:

**Source data 1.** Raw data for *Figure 6*.
**Figure supplement 1.** Trib3 as a possible link between CB$_1$R and C/EBP homologous protein (CHOP).
**Figure supplement 1—source data 1.**
**Figure supplement 2.** Hepatic ATF4 protein expression is regulated by CB$_1$R and is correlated with C/EBP homologous protein levels.
**Figure supplement 2—source data 1.** Raw data for Figure 6-Figure Supplement 2.

regulation in the expression levels of these proteins in our experimental paradigms (*Figure 2—figure supplement 1*), and the fact that we detected all changes in both mRNA and protein levels, we reasoned that the observed alterations in the circulating levels of sOb-R result from altered hepatic expression and secretion of the *Lepr* gene rather than decreased shedding. In fact, the regulation of ADAM10 and ADAM17 is complex and involves transcription, dynamic trafficking, cellular localization, and activity (*Edwards et al., 2008*; *Reiss and Bhakdi, 2017*). Whereas the current study is focused on the transcriptional regulation of sOb-R by CB$_1$R, other Ob-R isoforms, also expressed in humans, display (at the gene and protein levels) a trend similar to Ob-Re following either CB$_1$R activation or blockade. These isoforms (Ob-Ra, Ob-Rb) serve as substrates for ectodomain shedding; therefore, their transcriptional regulation may indirectly influence the sOb-R levels and be relevant to human physiology. In addition, numerous stimuli, such as activation of protein kinase C, an increase in intracellular calcium, lipotoxicity, and apoptosis, may contribute to the proteolytic cleavage of the extracellular leptin receptor domain (*Maamra et al., 2001*; *Schaab et al., 2012*). However, to the best of our knowledge, a molecular mechanism responsible for the decreased expression/shedding of sOb-R in obesity was never reported. Here we describe, for the first time, the involvement of the eCB/CB$_1$R system in regulating sOb-R levels and consequently leptin's activity.

The importance of the eCB/CB$_1$R system in regulating normal energy homeostasis as well as mediating obesity-related comorbidities is well acknowledged (review in *Simon and Cota, 2017*). In fact, its pivotal interaction with leptin has been first described in 2001, demonstrating that leptin reduces the content of hypothalamic eCBs (*Di Marzo et al., 2001*) and attenuates eCB-mediated 'retrograde' neuronal CB$_1$R signaling (*Jo et al., 2005*; *Malcher-Lopes et al., 2006*). On the other hand, activating CB$_1$R by eCBs may, in turn, regulate leptin levels and signaling, as suggested previously in women with anorexia nervosa, whose AEA levels are elevated (*Monteleone et al., 2005*), whereas their leptin levels are reduced. In fact, we have previously shown that CB$_1$R activation in adipocytes and pre-junctional sympathetic fibers innervating the adipose tissue stimulates leptin biosynthesis and release, and its activation in the proximal tubules of the kidney inhibits leptin degradation and renal clearance (*Tam et al., 2012*), thus possibly contributing to leptin resistance. In agreement with these findings, peripheral CB$_1$R blockade has been shown to ameliorate obesity-related hyperleptinemia, and subsequently restores leptin sensitivity in obese mice (*Tam et al., 2012*). By using both pharmacological and genetic approaches that target hepatic CB$_1$R, our findings here suggest another novel mechanism by which the eCB system may regulate hepatic leptin resistance. Specifically, peripheral blockade and hepatic deletion/overexpression of CB$_1$R modulate the expression levels of the sOb-R isoform in hepatocytes and its subsequent release into the circulation, reversing the CB$_1$R-mediated decrease in sOb-R levels and hepatic leptin resistance during obesity. One should point out that although liver-specific CB$_1$R KO mice retained higher levels of circulating sOb-R when fed a HFD, they were equally susceptible to DIO as their WT controls. Similarly, hepatic-specific CB$_1$R transgenic mice in the CB$_1$R KO background remained resistant to DIO while displayed significantly lower circulating sOb-R levels, as compared to their littermates. These data suggest that while liver CB$_1$R expression is a major contributor to circulating sOb-R levels, their roles in regulating systemic/central energy balance will need to be further validated. Nevertheless, the contribution of hepatic CB$_1$R to the regulation of hepatic leptin resistance was clearly demonstrated here by showing that DIO leads to a loss of leptin sensitivity in WT animals, but not in liver-specific CB$_1$R null

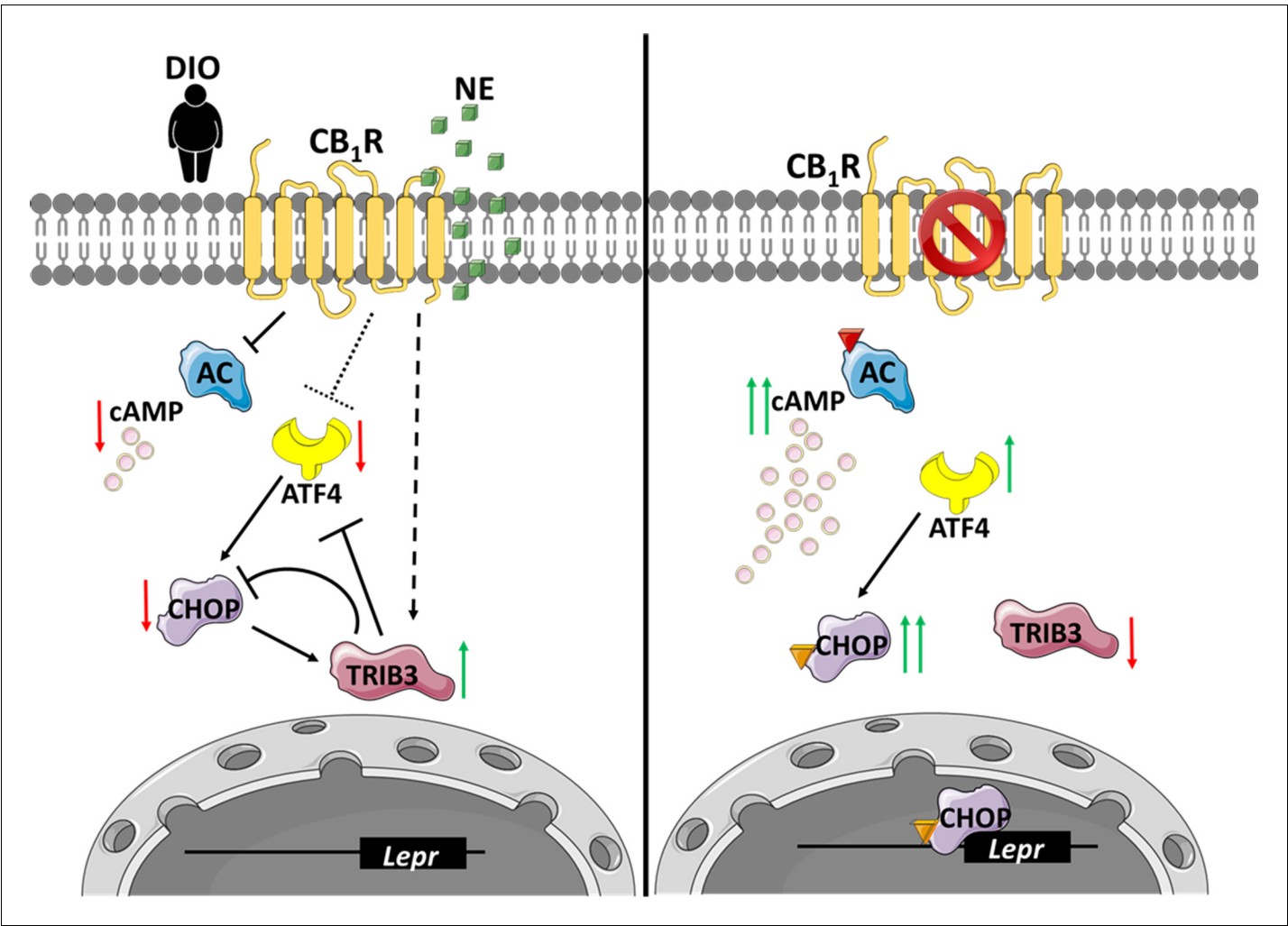

**Figure 7.** An illustration that describes the suggested molecular mechanism involving CB$_1$R and C/EBP homologous protein (CHOP) in the regulation of soluble isoform of leptin receptor (sOb-R) levels. (Left) When overexpressed in the liver, activated by endocannabinoids during diet-induced obesity (DIO) or synthetic cannabinoids, such as noladin ether (NE, green squares), CB$_1$R attenuates cAMP (pink circles) production by inhibiting adenylate cyclase (AC). CB$_1$R also represses the expression of ATF4, as well as upregulates Trib3 expression. As a consequence, CHOP levels are reduced, transcribing less *Lepr*. (Right) Blocking CB$_1$R in hepatocytes reverses these changes, leading to the activation and translocation of CHOP to the nucleus, which, in turn, directly binds the *Lepr* promoter and promotes the expression of sOb-R. Red arrows represent downregulation, whereas green arrows represent upregulation. Colored triangles represent activation.

obese mice. In line with these findings, overexpression of CB$_1$R in hepatocytes of lean mice inhibited leptin-induced STAT3 phosphorylation. These results, linking CB$_1$R with hepatic leptin signaling, may significantly advance our understanding of CB$_1$R's role in modulating hepatic gluconeogenesis and insulin sensitivity as well as lipid metabolism. Indeed, genetic deletion of CB$_1$R in hepatocytes partially protects mice from developing DIO-related hepatic steatosis, hyperglycemia, dyslipidemia, and insulin resistance (*Osei-Hyiaman et al., 2008*), whereas its overexpression in hepatocytes contributes to insulin resistance via inhibition of insulin signaling and clearance (*Liu et al., 2012*). In this sense, given that leptin regulates lipid, glucose, and insulin homeostasis in the liver, and that these metabolic functions are impaired in rodent models of increased eCB/CB$_1$R 'tone', a role of CB$_1$R-induced hepatic leptin resistance in regulating these processes can be postulated.

In accordance with our findings, Palomba and colleagues reported that CB$_1$R activation interferes with leptin's activity in hypothalamic ARC neurons (*Palomba et al., 2015*). On the other hand, opposite findings were reported by Bosier and colleagues, demonstrating that pharmacological or genetic deletion of CB$_1$R in astrocytes downregulates Ob-Rb expression and leptin-mediated

functional responses, whereas JZL195 (a dual MAGL and FAAH inhibitor) upregulates these features (*Bosier et al., 2013*). These differences can be explained by the distinct roles hepatocytes, astrocytes, and neurons play in peripheral and central metabolic regulations, and by cell-specific roles for $CB_1R$ in this regulation. As our findings demonstrate a similar effect of hepatic $CB_1R$ activation/over-expression or blockade/deletion on the different isoforms of LEPR, it seems equally possible that hepatic $CB_1R$ may affect DIO-related hepatic leptin resistance by not only modulating sOb-R levels, which controls leptin's activity, but also by modulating the expression of Ob-Rb in hepatocytes. Further investigations would allow us to differentiate between these two pathways.

Obesity is often characterized by an ER stress and consequently an adaptive unfolded protein response (UPR), operated by three parallel sensors: activating transcription factor 6 (ATF6), inositol requiring enzyme $1\alpha$ (IRE1$\alpha$), and protein kinase R-like ER kinase (PERK) (*Walter and Ron, 2011*). The activation of the latter induces the phosphorylation of eIF2$\alpha$, which, in turn, inhibits transcription and protein synthesis (*Ron, 2002*). In case of an extreme ER stress conditions, CHOP is activated by the PERK signaling pathway and executes ER stress-mediated apoptosis (*Hu et al., 2018*; *Zinszner et al., 1998*). In fact, ER stress has been shown to contribute to the development of hypothalamic leptin resistance, by impairing the transport of leptin across the BBB and suppressing STAT3 phosphorylation (*El-Haschimi et al., 2000*; *Hosoi et al., 2008*; *Ozcan et al., 2009*; *Zhang et al., 2008*). In addition, under physiological conditions, excess nutrients increases the demand for protein synthesis by the liver, leading to ER stress and UPR activation, which resolves the stress within hours (*Oyadomari et al., 2008*). Nevertheless, chronic ER stress in the liver was demonstrated in both obese mice and humans (*Ozcan et al., 2004*; *Puri et al., 2008*). Here we demonstrate that obese mice had elevated levels of phosphorylated eIF2$\alpha$, indicating increased ER stress, an effect that was reversed by peripheral $CB_1R$ blockade and was absent in LCB1 cKO. These findings are in agreement with our previous reports, where we reported that a neutral $CB_1R$ antagonist (AM6545) has the ability to reduce the HFD-induced upregulation in hepatic eIF2$\alpha$ (*Tam et al., 2010*), and that hepatic activation of $CB_1R$ induces ER stress and contributes to insulin resistance (*Liu et al., 2012*). Unexpectedly, we found that the hepatic gene and protein levels of CHOP and its upstream regulator ATF4 were significantly decreased in obese mice, and were upregulated by peripheral $CB_1R$ blockade. This observation, although counterintuitive, is conceptually in agreement with several previous reports that describe a modulated UPR signaling with altered sensitivity or output that might implicate conditions of persistence/repeated stress (*Chambers et al., 2012*; *Gomez and Rutkowski, 2016*; *Preissler et al., 2015*; *Yang et al., 2015*).

Apart from its role in ER stress-mediated apoptosis, CHOP has been implicated in regulating other processes such as inflammation (*Endo et al., 2006*; *Nakayama et al., 2010*), insulin resistance (*Maris et al., 2012*; *Song et al., 2008*), and adiposity. Specifically in the liver, Chikka and colleagues suggest that CHOP is a suppressor of key regulators of lipid metabolism like *Cebpa*, *Ppara*, and *Srebf1* (*Chikka et al., 2013*), and demonstrate that CHOP-deficient mice tend to develop hepatic steatosis in response to bortezomib-induced ER stress. This is in agreement with an earlier report describing higher body weight and adiposity in female CHOP KO mice compared to WT controls (*Ariyama et al., 2007*). In contrast, we show that male CHOP KO mice and their WT littermate controls gain comparable amount of weight and have similar body composition following exposure to an HFD for 14 weeks. Nevertheless, we did see a trend toward increased liver triglycerides.

An interesting observation was that the eCB 'tone' of lean and obese CHOP KO mice was comparable. To the best of our knowledge, a direct regulation of eCB synthesis or degradation by CHOP has never been described. Thus, our data imply a possible link between the two. In fact, it is possible that the higher basal eCB levels seen in CHOP KO mice in comparison with their littermate controls are the consequence of leptin's reduced ability to inhibit AEA and 2-AG production. This may be due to the reduced level of sOb-R found in the liver of CHOP KO animals. This hypothesis, although not tested here and which needs further experimental corroboration, is in accordance with the findings of others who demonstrated such a mechanism in the hypothalamus and in adipocytes treated with leptin (*Di Marzo et al., 2001*; *Matias et al., 2006*). In addition, JD5037 failed to reverse many of the metabolic abnormalities, such as HDL and LDL content as well as liver triglycerides in DIO CHOP KO mice. It also had much smaller effect on total body fat mass then in WT DIO mice, suggesting an obligatory role of CHOP in mediating the metabolic improvements induced by $CB_1R$ blockade. Whereas basal circulating levels of sOb-R were comparable between WT and CHOP KO mice, its levels were markedly lower in the liver of CHOP KO mice as well as cultured hepatocytes.

Moreover, sOb-R remained low in these mice even on HFD, supporting a role for CHOP in regulating the synthesis of sOb-R in the liver. The failure of JD5037 to elevate sOb-R levels in obese CHOP KO mice places CHOP downstream of $CB_1R$ in this molecular cascade.

As mentioned earlier, the molecular signaling pathway(s) by which eCBs or $CB_1R$ regulates CHOP levels is outside the scope of this work. However, two possible mechanisms might be relevant. The first putative mechanism may involve Trib3, which is known to be induced in a broad range of cells and in response to multiple forms of cellular stress such as ER stress, excess of free fatty acids, oxidative stress, hypoxia, hyperglycemia, and toxins (reviewed in *Ord and Ord, 2017*). Interestingly, it has been shown that $\Delta^9$-THC as well as synthetic cannabinoid agonists upregulate Trib3 expression (*Blázquez et al., 2006*; *Carracedo et al., 2006*; *Salazar et al., 2013*; *Vara et al., 2011*) to engage apoptosis in variable cancer models. Moreover, Cinar et al. demonstrated that hepatic $CB_1R$ induces ER stress in hepatocytes by increasing de novo synthesis of ceramides (*Cinar et al., 2014*), which are also involved in Trib3 upregulation following ER stress (*Carracedo et al., 2006*). In line with our observation that Trib3 levels are negatively correlated with ATF4, CHOP, and sOb-R and are elevated following direct or indirect activation of $CB_1R$, we suggest that Trib3 is a molecular linker between $CB_1R$ and the ATF4/CHOP complex. In fact, Trib3 directly interacts with and inhibits ATF4 and CHOP, forming a negative feedback on the regulation of their activity (*Ohoka et al., 2005*). This Trib3-induced negative modulation of ATF4 and CHOP has been suggested to contribute to the fine-tuning of ATF4- and CHOP-dependent transcription in stressed cells, such as hepatocytes exposed to fatty acid flux. With the extensive body of evidence demonstrating a wide range of molecules that are known to regulate Trib3 expression, our current findings highlight $CB_1R$ as a novel possible modulator that regulates Trib3 transcription, thus suggesting that $CB_1R$ activation may disrupt the ER stress signaling pathway involving eIF2$\alpha$, ATF4, and CHOP. However, the molecular events that link $CB_1R$ and Trib3 require further assessment, and direct regulation of ATF4 by $CB_1R$ cannot be excluded. Second possible mechanism has to do with CHOP being a cAMP responsive protein, so its expression is induced via a cAMP response element (CRE) (*Conkright et al., 2003*; *Pomerance et al., 2003*; *Ramji and Foka, 2002*; *Wilson and Roesler, 2002*). $CB_1R$, a G-protein coupled receptor (GPCR), which upon activation recruits Gi protein, can inhibit the activity of adenylyl cyclase and reduce the levels of cAMP (*Turu and Hunyady, 2010*). It is therefore plausible that a decline in cAMP following $CB_1R$ activation inhibits CHOP transcription. This hypothesis is more appealing if one considers the pivotal role of cAMP in regulating liver metabolism (*Wahlang et al., 2019*), and takes into account the fact that reduced levels of cAMP were documented in HFD-fed mice (*Zingg et al., 2017*). Yet, further studies will need to explore the specific molecular pathways linking together hepatic eCB/$CB_1R$ system and CHOP.

In conclusion, we report a new role for the hepatic eCB/$CB_1R$ in the development of hepatic leptin resistance, by reducing the expression and/or subsequent release of sOb-R (*Figure 7*). We show that peripherally restricted $CB_1R$ antagonism has the ability to restore sOb-R levels, contributing to the reversal of obesity-induced hyperleptinemia. We also suggest that upon $CB_1R$ blockade in hepatocytes, ATF4 as well as CHOP levels are upregulated via reduced Trib3 expression. CHOP, in turn, directly binds the LEPR promoter and promotes the expression of sOb-R.

## Materials and methods

### Key resources table

| Reagent type (species) or resource | Designation | Source or reference | Identifiers | Additional information |
|---|---|---|---|---|
| Gene (*Mus musculus*) | *Lepr* | e!Ensembl | ENSMUS G00000057722 | Leptin receptor |
| Strain, strain background (*Mus musculus*) | Wild type | Envigo Israel | C57Bl/6N | |
| Strain, strain background (*Mus musculus*) | LCB1 cKO | *Osei-Hyiaman et al., 2008* | | Liver-specific CB1R KO |

*Continued on next page*

*Continued*

| Reagent type (species) or resource | Designation | Source or reference | Identifiers | Additional information |
|---|---|---|---|---|
| Strain, strain background (*Mus musculus*) | hTgCB1 cKO | *Tam et al., 2010* | | CB1R KO, overexpressing CB1R in hepatocytes |
| Strain, strain background (*Mus musculus*) | CHOP KO | The Jackson Laboratory | B6.129S(Cg)-Ddit3[tm2.1Dron]/J, #005530 RRID:IMSR_JAX:005530 | |
| Transfected construct (*Mus musculus*) | mCHOP-9E10 | Addgene | CHOP6: mCHOP-WT-9E10-pCDNA, #21913 RRID:Addgene_21913 | |
| Transfected construct (Firefly,sv40) | pGL3 | Promega | pGL3-basic vector, E1751 | Luciferase Assay vector |
| Transfected construct (*Mus musculus*) | Ad-GFP-mLEPR | VECTOR BIOSYSTEMS Inc | ADV-263380 | |
| Cell line (*Mus musculus*) | Wild type | *Uzi et al., 2013* | | Immortalized mouse hepatocytes |
| Cell line (*Mus musculus*) | CHOP KO | *Uzi et al., 2013* | | Immortalized mouse hepatocytes |
| Antibody | LEPR (rabbit polyclonal) | Novus | NB-120–5593 RRID:AB_791038 | WB (1:2000) |
| Antibody | pSTAT3 (rabbit monoclonal) | Cell signaling | #9145 RRID:AB_2491009 | Phosphorylated Stat3 (Tyr705), WB (1:1000) |
| Antibody | STAT3 (mouse monoclonal) | Cell signaling | #9139 RRID:AB_331757 | WB (1:3000) |
| Antibody | p-eIF2α (rabbit polyclonal) | Cell signaling | #9721 RRID:AB_330951 | Phosphorylated eIF2α (Ser51), WB (1:1000) |
| Antibody | t-eIF2α (rabbit polyclonal) | Cell signaling | #9722 RRID:AB_2230924 | Total eIF2α WB (1:1000) |
| Antibody | ATF4 (rabbit monoclonal) | Cell signaling | #11815 RRID:AB_2616025 | WB (1:1000) |
| Antibody | CHOP (mouse monoclonal) | Cell signaling | #2895S RRID:AB_2089254 | WB (1:1000) ChIP (2.5 µg/sample) |
| Antibody | Trib3 (rabbit polyclonal) | Abcam | ab137526 RRID:AB_2876352 | WB (1:2000) |
| Antibody | β-Actin (mouse monoclonal) | Abcam | ab49900 RRID:AB_867494 | WB (1:30,000) |
| Antibody | H3 (rabbit polyclonal) | Abcam | ab1791 RRID:AB_302613 | ChIP (2.5 µg/sample) |
| Antibody | Anti-rabbit HRP (donkey polyclonal) | Abcam | ab97085 RRID:AB_10679957 | WB (1:10,000) |
| Antibody | Anti-mouse HRP (donkey polyclonal) | Abcam | ab98799 RRID:AB_10675068 | WB (1:10,000) |
| Commercial assay or kit | SLR ELISA | Shanghai Bluegene Biotech | E03S0226 | |
| Commercial assay or kit | Triglyceride Assay Kit | Abcam | ab65336 | |
| Commercial assay or kit | Dual-Glo Luciferase Assay System | Promega | E2920 | |
| Chemical compound, drug | JD5037 | Haoyuan Chemexpress Co., Ltd | HY-18697 | |

*Continued on next page*

*Continued*

| Reagent type (species) or resource | Designation | Source or reference | Identifiers | Additional information |
|---|---|---|---|---|
| Chemical compound, drug | NE | Cayman Chemicals | 62165 | 2-Arachidonyl glycerol ether |
| Chemical compound, drug | Tunicamycin (TM) | Holland Moran | 11089-65-9 | |
| Software, algorithm | GraphPad Prism | GraphPad Software | RRID:SCR_002798 | |

## Animals and experimental protocol

All animal studies were approved by the Institutional Animal Care and Use Committee of the Hebrew University of Jerusalem (AAALAC accreditation #1285; Ethic approval numbers MD-14–14008 and MD-19–15951). Animal studies are reported in compliance with the ARRIVE guidelines (*NC3Rs Reporting Guidelines Working Group et al., 2010*), and are based on the rule of the replacement, refinement, or reduction. All the animals used in this study were housed under specific pathogen-free (SPF) conditions, up to five per cage, in standard plastic cages with natural soft sawdust as bedding. The animals were maintained under controlled temperature of 22–24°C, humidity at 55 ± 5%, and alternating 12 hr light/dark cycles (lights were on between 7:00 and 19:00 hr), and provided with food and water ad libitum. C57Bl/6 (Envigo, Israel), LCB1 cKO, and hTgCB1 cKO (kindly provided by Dr. George Kunos, NIH) or B6.129S(Cg)-Ddit3$^{tm2.1Dron}$/J (CHOP KO, The Jackson Laboratory #005530), and their WT littermate controls were used for in vivo experiments. All mice were male and 8–10 weeks old at the beginning of each experiment. To generate DIO (body weight >42 g), mice were fed with a standard diet (STD; 14% Kcal fat, 24% Kcal protein, 62% Kcal carbohydrates; NIH-31 rodent diet) or a HFD (60% Kcal fat, 20% Kcal protein, and 20% Kcal carbohydrates; Research Diet, D12492) for 14 weeks. Then, obese mice were randomly divided into the experimental groups. Treatment with JD5037 (3 mg/kg, ip) or vehicle (1% Tween80, 4% DMSO, 95% Saline) was conducted for 7 days, and 12 hr following the last dose, the mice were euthanized by a cervical dislocation under anesthesia, and blood and livers were harvested for further analyses. For leptin sensitivity test, mice were fasted for 24 hr before an ip administration of recombinant mouse leptin (3 mg/kg). One hour following leptin administration, mice were euthanized and livers were harvested and processed for phosphorylated STAT3 detection using western blot.

## Cell culture

WT or CHOP KO immortalized hepatocytes (described in *Uzi et al., 2013*), confirmed to be mycoplasma-negative, were maintained in DMEM (01-100-1A; Biological Industries, Israel) supplemented with 5% FCS, 100 mM glutamine, 100 mM Na-Pyruvate, and Pen/Strep. Cells were cultured at 37°C in a humidified atmosphere of 5% $CO_2$/95% air. To test the effect of $CB_1R$ activation, cells were seeded in 6-well plates (25 × 10$^4$ cells/well) for 24 hr. Then, growth medium was replaced with a serum-free medium (SFM) for an additional 12 hr. At the morning of the experiment the medium was replaced with fresh SFM containing either vehicle (EtOH), 2.5 μM NE (Cayman Chemicals, Ann Arbor, Michigan) or a combination of 100 nM JD5037 (Haoyuan Chemexpress Co., Ltd) and 2.5 μM NE. After 24 hr, cells were harvested for further analyses as described below.

## Measurements of sOb-R

Soluble leptin receptor was quantified by an ELISA kit, capable to differentiate the soluble isoform from other isoforms, according to manufacturer's instructions (E03S0226; Shanghai Bluegene Biotech, China). Briefly, for serum measurements, we diluted serum in saline (1:2) and 100 μL from the diluted samples were analyzed. For hepatic measurements, 50–100 mg tissue samples were homogenized in 300 μL of 1× PBS and centrifuged for 5 min in 5000 rpm; 100 μL of cleared lysates were analyzed. Data were normalized to sample protein content, determined using the Pierce BCA Protein Assay Kit (Thermo Scientific, IL).

To measure sOb-R protein content in cell culture media, we used trichloroacetic acid (TCA) precipitation protocol; 350 μL of 100% TCA were added to 1.6 mL culture media, vortexed, and incubated for 30 min on ice. Samples were then centrifuged to pellet proteins (14,000 rpm, 10 min, 4°C).

Pellets were washed in 100% acetone, resuspended in 0.1 M NaOH and protein loading dye, and analyzed by western blot. Ponceau staining of the blots was used as loading control for quantification.

Validation of LEPR antibody. The specificity of the anti-LEPR antibody was validated in a control experiment (*Figure 1—figure supplement 1*), where mouse LEPR was overexpressed in kidney cell line by using a viral infection. The viral vector encoded Ad-GFP-mLEPR (ADV-263380, VECTOR BIO-SYSTEMS Inc) was used in a multiplicity of infection of 50, and cells were harvested for western blot analysis 24 hr post infection.

## Real-time PCR

For total mRNA isolation, tissue samples or hepatocytes were washed in 1× PBS and harvested using Bio-Tri RNA lysis buffer (Bio-Lab, Israel). Extracted RNA was treated with DNase I (Thermo Scientific, IL), and reverse transcribed using the Iscript cDNA kit (Bio-Rad Laboratories, CA). Quantitative PCR reactions for *Lepr-s*, *Ddit3*, or *Ppp1r15a* were performed using iTaq Universal SYBR Green Supermix (Bio-Rad Laboratories, CA), and the CFX connect ST system (Bio-Rad Laboratories, CA). Relative quantity (RQ) values of all tested genes were normalized to *Ubc*. Primers are listed in *Supplementary file 1*.

## Western blot analysis

Tissue samples or hepatocytes were washed in cold 1× PBS, and harvested in a RIPA buffer (25 mM Tris-HCl pH 7.6, 150 mM NaCl, 1% NP-40, 1% sodium deoxycholate, 0.1% SDS), vortexed and incubated for 30 min at 4°C, and then centrifuged for 10 min at 14,000 rpm. Protein concentrations were determined using the Pierce BCA Protein Assay Kit (Thermo Scientific, IL). Cleared lysates were supplemented with protein sample buffer, resolved by SDS-PAGE (4–15% acrylamide, 150 V) and transferred to PVDF membranes using the Trans-Blot Turbo Transfer System (Bio-Rad Laboratories, CA). Membranes were incubated for 1 hr in 5% milk (in TBS-T) to block unspecific binding, washed briefly, and incubated overnight at 4°C with the following primary antibodies: LEPR (NB-120–5593, Novus), phosphorylated STAT3 (9145, Cell Signaling), STAT3 (9139, Cell Signaling), phosphorylated eIF2α (9721, Cell Signaling), eIF2α (9722, Cell Signaling), ATF4 (11815, Cell Signaling), CHOP (2895S, Cell Signaling), Trib3 (ab137526, Abcam), or β-Actin (ab49900, Abcam). Anti-rabbit (ab97085, Abcam) or mouse (ab98799, Abcam) horseradish peroxidase (HRP)-conjugated secondary antibodies were used for 1 hr at room temperature, followed by chemiluminescence detection using Clarity Western ECL Blotting Substrate (Bio-Rad Laboratories, CA). Densitometry was quantified using ImageLab software. Protein RQ was calculated as the ratio between LEPR to total protein signal (ponceau) in culture media supernatants and to β-actin in cell and tissue lysates.

## Body composition and biochemical analysis

Total body fat and lean masses were determined by EchoMRI-100H (Echo Medical Systems LLC, Houston, TX, USA).

HDL and LDL measurements were done using the Cobas C-111 chemistry analyzer (Roche, Switzerland).

## Hepatic triglycerides measurements

Tissue lipids were extracted as described in *Folch et al., 1957*, and quantified using Triglyceride Assay Kit (ab65336; Abcam). Data were normalized to tissue weight.

## eCB measurements by LC-MS/MS

eCBs were extracted, purified, and quantified in liver homogenates, as described previously (*Drori et al., 2019*; *Udi et al., 2017*). LC-MS/MS was analyzed on an AB Sciex (Framingham, MA, USA) Triple Quad 5500 mass spectrometer coupled with a Shimadzu (Kyoto, Japan) UHPLC System. eCBs were detected in a positive ion mode using electron spray ionization (ESI) and the multiple reaction monitoring (MRM) mode of acquisition. The levels of each compound were analyzed by monitoring multiple reactions. The molecular ion and fragment for each compound were measured as follows: m/z 348.3→62.1 (quantifier) and 91.1 (qualifier) for AEA, m/z 379.3→287.3 (quantifier)

and 91.1 (qualifier) for 2-AG. The levels of AEA and 2-AG in samples were measured against standard curves and normalized to tissue weight.

## Chop overexpression

WT hepatocytes were transfected with mCHOP-WT-9E10-pCDNA1 vector (Addgene plasmid #21913) using Lipofectamin 3000. Cells were harvested 24 hr post-transfection and CHOP expression was validated by western blot analysis.

## Luciferase promoter assay

*Mus musculus* Ob-R promoter sequence (GRCm38:CM000997.2. Chromosome 4: 101,716,750–101,718,250 forward strand) was cloned into pGL3-basic vector (E1751, Promega). Reporter vector and Renilla luciferase vector were then co-transfected into WT or CHOP KO hepatocytes using Lipophectamin 3000. Twenty-four hours post transfection, cells were treated with either vehicle, 2.5 µg/mL tunicamycin (11089-65-9; Holland Moran, Israel) or 2.5 µM NE for indicated period. At the end of the experiment, luciferase activity was measured using Dual-Glo Luciferase Assay System (E2920, Promega). Data are presented as the ratio between Firefly and Renilla luciferase activity.

## Chromatin immunoprecipitation

WT or CHOP KO hepatocytes were seeded in 100 mm plate and left to adhere ($5 \times 10^6$ cells per plate; three plates for each sample). The next day, cells were treated with either DMSO or 2.5 µg/mL tunicamycin for 6 hr to induce CHOP expression. At the end of incubation, cells were washed in $1\times$ PBS, fixed with 1% formaldehyde for 10 min, then quenched with 125 mM glycine and harvested from plates. Following centrifugation, cells were resuspended in a lysis buffer and sonicated for 12 cycles of 30 s pulse followed by 30 s rest in 70% amplitude. Sheared DNA was diluted in a ChIP dilution buffer and pre-cleared with magnetic ProteinG-sepharose beads for 4 hr at 4˚C. Ten percent of the lysate was removed and saved as 'Input'. The rest of the lysate was divided and each part was incubated overnight at 4˚C with 2.5 µg of either anti-H3 (ab1791, Abcam), anti-CHOP (2895S, Cell Signaling), or IgG isotype control. Antibody-chromatin complexes were precipitated with magnetic protein G-Sepharose beads, washed with low salt, high salt, lithium chloride, and Tris-EDTA buffers. DNA was then eluted from beads, digested with proteinase K, and purified; 1.5 µL of clean DNA was used in a qPCR reaction using specific primers for *Gapdh* or *Lepr* promoter region. For a positive control, *Ppp1r15a* primers were used. For each sample, we calculated the ratio between the RQ (expressed as % of input) of ObR promoter qPCR product in αCHOP IP relative to αH3 IP. This ratio was normalized to the ratio of *Gapdh* qPCR product to control for nonspecific binding. Data are expressed as the fold-change of this ratio in tunicamycin-treated compared to vehicle-treated cells. ChIP primers are listed in *Supplementary file 1*.

## Statistics

The data and statistical analysis comply with the recommendations on experimental design and analysis as reported previously (*Curtis et al., 2018*). Randomization was used to assign samples to the experimental groups and treatment conditions for all in vivo studies. Data collection and acquisition of all in vivo and in vitro experimental paradigms were performed in a blinded manner. Data are presented as mean ± SEM. Unpaired two-tailed Student's t-test was used to determine variations between two groups. Results in multiple groups were compared by ANOVA followed by a Bonferroni post hoc analysis using GraphPadPrism v6 for Windows. Post hoc tests were conducted only if *F* was significant, and there was no variance inhomogeneity. Significance was set at $p < 0.05$.

## Acknowledgements

This study was supported by grants from the Israeli Science Foundation (ISF; 617/14 and 158/18), The Obesity Society's Early Career Research Award, and an ERC-2015-StG grant (#676841) to J.T, as well as the Hungarian National Research, Development, and Innovation Office (Grant #NKFI-6/FK_124038 to G.S.). This paper is dedicated to the memory of Simon Tam.

## Additional information

### Funding

| Funder | Grant reference number | Author |
|---|---|---|
| Israel Science Foundation | 617/14 | Joseph Tam |
| Israel Science Foundation | 158/18 | Joseph Tam |
| Obesity Society | Early Career Research Award | Joseph Tam |
| ERC-2015-StG grant | 676841 | Joseph Tam |
| Hungarian National Research, Development, and Innovation Office | NKFI-6/FK_124038 | Gergő Szanda |

The funders had no role in study design, data collection and interpretation, or the decision to submit the work for publication.

### Author contributions

Adi Drori, Data curation, Formal analysis, Validation, Investigation, Methodology, Writing - original draft; Asaad Gammal, Daniel Wesley, Investigation; Shahar Azar, Liad Hinden, Data curation, Investigation; Rivka Hadar, Investigation, Project administration; Alina Nemirovski, Methodology; Gergő Szanda, Resources, Writing - review and editing; Maayan Salton, Resources, Data curation, Formal analysis, Methodology, Writing - review and editing; Boaz Tirosh, Resources, Formal analysis, Methodology, Writing - review and editing; Joseph Tam, Conceptualization, Formal analysis, Supervision, Funding acquisition, Validation, Investigation, Visualization, Writing - original draft, Writing - review and editing

### Author ORCIDs

Adi Drori https://orcid.org/0000-0002-5790-9544
Liad Hinden http://orcid.org/0000-0002-0307-4350
Boaz Tirosh http://orcid.org/0000-0001-8067-6577
Joseph Tam https://orcid.org/0000-0002-0948-0093

### Ethics

Animal experimentation: All animal studies were approved by the Institutional Animal Care and Use Committee of the Hebrew University of Jerusalem (AAALAC accreditation #1285; Ethic approval numbers MD-14-14008 & MD-19-15951). Animal studies are reported in compliance with the ARRIVE guidelines (Kilkenny et al., 2010), and are based on the rule of the replacement, refinement, or reduction. All the animals used in this study were housed under specific pathogen-free (SPF) conditions, up to five per cage, in standard plastic cages with natural soft sawdust as bedding. The animals were maintained under controlled temperature of 22-24°C, humidity at 55 ± 5%, and alternating 12-hour light/dark cycles (lights were on between 7:00 and 19:00 hr), and provided with food and water ad libitum.

### Decision letter and Author response

Decision letter https://doi.org/10.7554/eLife.60771.sa1
Author response https://doi.org/10.7554/eLife.60771.sa2

## Additional files

### Supplementary files

- Supplementary file 1. Supplementary Tables 1 and 2.
- Transparent reporting form

## Data availability

All data generated or analysed during this study are included in the manuscript and supporting files. Source data files have been provided for all figures.

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
