## [Decision Letter]

Thank you for submitting your article "CB_1_R Regulates Soluble Leptin Receptor Levels via CHOP, Contributing to Hepatic Leptin Resistance" for consideration by *eLife*. Your article has been reviewed by three peer reviewers, and the evaluation has been overseen by a Reviewing Editor and David James as the Senior Editor. The following individuals involved in review of your submission have agreed to reveal their identity: George Kunos (Reviewer #1); Vincenzo Di Marzo (Reviewer #2); Thomas Scherer (Reviewer #3).

The reviewers have discussed the reviews with one another and the Reviewing Editor has drafted this decision to help you prepare a revised submission.

Summary:

Leptin resistance is a major pathogenic factor in obesity and both central and peripheral mechanisms have been implicated in its development. Circulating leptin bound to a short form of soluble leptin receptor (sOb-R encoded by Ob-Re) cannot bind to the functional leptin receptor, resulting in reduced levels of free leptin that can cross the blood-brain-barrier. Circulating levels of sOb-R are inversely related to plasma leptin levels, suggesting that they play a key role in the control of leptin sensitivity. Activation of the endocannabinoid/CB_1_ receptor system is known to mediate obesity-related leptin resistance in rodents, and Tam and co-workers present elegant evidence for a novel mechanism by which endocannabinoids acting via hepatocyte CB_1_R down-regulate the hepatic expression and circulating levels of sOb-R, thus contributing to leptin resistance.

Reviewer 1:

1) Pharmacological (CB_1_ agonist/antagonist) as well as genetic tools (liver-specific CB_1_ ko and ki) are used to convincingly demonstrate that CB_1_R activation downregulates Ob-Re mRNA and sOb-R protein in the liver and in the circulation. Surprisingly, CB_1_R signal via reducing the expression of the pro-apoptotic transcription factor CHOP, an integral component of the PERK/eIF2α/ATF4/CHOP ER-stress pathway, as supported through the use of CHOP-/- mice. While the evidence is convincing, it is counterintuitive. ER-stress induced by a high fat diet was reported to upregulate rather than reduce CHOP expression in the liver (e.g. Bae et al., Molecules 25:2667, 2020) and the authors' own finding of increased eIF2α phosphorylation, which is upstream from CHOP, should result in increased rather than decreased CHOP expression. Could CB_1_ receptor activation disrupt this pathway at a point between eIF2α and CHOP? Could a Trib3-mediated decrease in ATF4 phosphorylation represent such a 'break'? This could be tested by monitoring ATF4 phosphorylation in the authors' paradigm. At a minimum, this paradox needs to be discussed.

2) In the subsection “CHOP contributes to the metabolic response to peripheral CB_1_R blockade”, the loss of ability of the CB_1_ antagonist to reverse the CB_1_-induced dyslipidemic effects (Figures 3H, I, J) in CHOP-/- mice is attributed to an increase in 'endocannabinoid tone', or more specifically a ~2-3-fold increase in the tissue levels of the endocannabinoids anandamide and 2-AG. In the absence of dose-response data, there is no reason to believe that a modest increase in the local concentration of endogenous agonists would result in the loss of effect of a highly potent competitive antagonist. It is more likely that the genetic deletion of an obligatory downstream mediator (CHOP) of the CB_1_ response is responsible for the loss of the effect of the antagonist. In this case the unchanged response to HFD may be mediated by a non-CB_1_ mechanism.

3) A limitation of the impact of these findings is that they may not be extended to human physiology, as in humans circulating sOb-R is generated exclusively by ectodomain shedding catalyzed by the proteolytic enzymes ADAM10 and ADAM17, whose expression remained unchanged (Discussion). However, only mRNA was measured, which does not exclude possible changes in enzyme protein levels or enzyme activity. This limitation needs to be mentioned/discussed.

4) The authors demonstrate that obesity leads to loss of leptin sensitivity in the wild-type, but not in LCB1ko liver, as indicated by the loss of leptin-induced STAT3 phosphorylation in the former but not the latter. These important findings should be more prominently highlighted in the Discussion, which should also briefly touch on what's known about the effects of leptin in the liver (e.g. regulation of hepatic insulin sensitivity).

Reviewer 2:

1) There are points in the Introduction and Discussion that need however to be ameliorated, through minor but nevertheless important corrections, as detailed below:

a) In general, the reader may wonder why the authors decided to investigate the role of CHOP in the effects of CB_1_R on sObR levels. This should be very briefly mentioned in the Introduction.

b) In the Introduction, the authors mention that the eCB system is overactive in obesity without quoting one of the first paper showing that this is true not only in the hypothalamus but also in peripheral cells, including the major source of leptin, i.e. the adipocytes (Matias et al., 2006). This should be mentioned; additionally, Monteleone et al., 2005, showed probably for the first time the elevation of circulating anandamide levels in overweight/obese women with binge eating disorder, again to be mentioned here;

c) In the Discussion, when referring to leptin signaling in anorexia nervosa, the authors should again mention the Monteleone et al., 2005 study, showing how anandamide levels are also elevated in women with this disorder, possibly due to their low leptin levels;

d) the above mentioned study by Matias et al., 2006, showed how both acute and chronic leptin significantly downregulates both anandamide and 2-AG levels in adipocytes (Figure 1 of that paper). This finding could be related to, or even explain, the authors' finding of higher eCB levels in CHOP-KO mice, which, by producing much less sObR, would counteract this inhibition of eCB levels also in adipocytes, by disinhibiting it. Since the authors did not provide a molecular mechanism for this effect, they may wish to discuss this possibility, especially if they have data showing that also in hepatocytes, like in adipocytes and hypothalamus, leptin decreases endocannabinoid levels via sObR (but also if they don't have such data).

Reviewer 3:

1) From a methodological viewpoint the study is well executed, however what I am missing a bit is the discussion of the physiologic significance and relevance of the findings for the field. This is relevant, because the exact role of the soluble leptin receptor for leptin action is not fully resolved in the current literature. Since the liver-specific CB_1_R transgenic mice on CB_1_R-null background are resistant to diet induced obesity, despite lower soluble leptin receptor expression raises the question whether the observed phenomenon is relevant for overall leptin action, which is predominantly mediated via the CNS. In the CB_1_R transgenic mice central leptin action seems (based on the resistance to DIO) preserved or at least not majorly impaired, despite lower secretion of soluble leptin receptor, questioning the overall physiologic relevance of the observed reduction in soluble leptin receptor for whole body leptin action. Do the authors have additional data on leptin signaling (in liver and brain) in the transgenic mice?

2) The claim that the regulation of soluble leptin receptor secretion via liver CB_1_R directly relates to liver leptin resistance is in my opinion not fully supported. The authors only present STAT3 phospho western blot data in CB_1_ KO and WT mice, but not the CB_1_R transgenic/CHOP KO mice. Direct effects of the endocannabinoid system on STAT3 signaling can thus not be excluded. I suggest deleting the claim from the title/Abstract. The authors should rather expand on this theory in the Discussion.

3) Circulating leptin levels (Figure 4F) in CHOP KO vs. WT JD5037 treated mice on HFD seem to correlate with fat mass (Figure 3F) rather than soluble leptin receptor levels. Are there any correlation data that would support the claim that soluble leptin receptor levels directly affects circulating leptin levels?

---

## [Author Response]

Reviewer 1:1) Pharmacological (CB_1_ agonist/antagonist) as well as genetic tools (liver-specific CB_1_ ko and ki) are used to convincingly demonstrate that CB_1_R activation downregulates Ob-Re mRNA and sOb-R protein in the liver and in the circulation. Surprisingly, CB_1_R signal via reducing the expression of the pro-apoptotic transcription factor CHOP, an integral component of the PERK/eIF2α/ATF4/CHOP ER-stress pathway, as supported through the use of CHOP-/- mice. While the evidence is convincing, it is counterintuitive. ER-stress induced by a high fat diet was reported to upregulate rather than reduce CHOP expression in the liver (e.g. Bae et al., Molecules 25:2667, 2020) and the authors' own finding of increased eIF2α phosphorylation, which is upstream from CHOP, should result in increased rather than decreased CHOP expression. Could CB_1_ receptor activation disrupt this pathway at a point between eIF2α and CHOP? Could a Trib3-mediated decrease in ATF4 phosphorylation represent such a 'break'? This could be tested by monitoring ATF4 phosphorylation in the authors' paradigm. At a minimum, this paradox needs to be discussed.

We would like to thank the reviewer for finding our data convincing and important.

As stated by the reviewer, high-fat diet (HFD)-induced obesity is known to upregulate ER-stress in the liver. We have previously demonstrated it (Tam et al., 2010) and are currently reporting it here by showing the increased hepatic phosphorylation of eIF2α (Figure 3—figure supplement 1). Indeed, the reduced transcriptional expression levels of CHOP are counterintuitive. Yet, as suggested by the reviewer, this could be explained via a CB_1_R-dependent regulation of either ATF4 or Trib3. The latter is upregulated in response to diverse forms of cellular stress (such as ERstress, excess free fatty acid, oxidative stress, hypoxia, hyperglycemia, and toxins; reviewed in Örd and Örd, 2017), and also by cannabinoids. Trib3 is able to directly bind and inhibit ATF4 and CHOP. In fact, our data support such a molecular mechanism, since both HFD and CB_1_R activation upregulate the mRNA and protein expression of Trib3 in the liver/hepatocytes and peripheral CB_1_R blockade in HFD-induced obese mice normalizes its expression (Figure 6—figure supplement 1).

Following the comment raised by the reviewer, we expanded our evidence for such a regulation and assessed the hepatic mRNA and protein expression levels of ATF4 in three animal settings: WT mice (STD- and HFD-fed animals treated with/without the peripherally restricted CB_1_R blocker, JD5037), liver-specific CB_1_ null mice (LCB1 KO, STD vs. HFD), and animals with overexpression of CB_1_R only in hepatocytes (htgCB1^-/-^ mice, fed STD, or HFD). As can be seen in the new Figure 6—figure supplement 2, the hepatic ATF4 expression is downregulated in DIO mice, an effect that was completely normalized by peripheral CB_1_R blockade. Also, whereas nullification of CB_1_R in hepatocytes prevented the HFD-induced downregulation of ATF4, overexpression of the former in hepatocytes significantly contributes to the reduced expression of the latter. In addition, preliminary studies conducted in our lab revealed that direct CB_1_R activation by AEA in primary mouse hepatocytes results in a time-dependent reduction in ATF4 and CHOP mRNA levels (please see Author response image 1).

**Author response image 1. sa2fig1:** 

Taken together, these findings support the idea that although ER stress is induced in the liver by HFD, the activation/blockade (or genetic deletion) of CB_1_R disrupts the conventional unfolded protein response (UPR) by affecting ATF4 expression, either directly, or indirectly via Trib3. These new findings are now described in the subsection “CHOP plays a key role in the regulation of sOb-R by the eCB/CB_1_R system”, and are further discussed in the Discussion. We have also amended the illustration that describes the molecular mechanism by which CB_1_R regulates sOb-R levels (please see the revised Figure 7).

2) In the subsection “CHOP contributes to the metabolic response to peripheral CB_1_R blockade”, the loss of ability of the CB_1_ antagonist to reverse the CB_1_-induced dyslipidemic effects (Figures 3H, I, J) in CHOP-/- mice is attributed to an increase in 'endocannabinoid tone', or more specifically a ~2-3-fold increase in the tissue levels of the endocannabinoids anandamide and 2-AG. In the absence of dose-response data, there is no reason to believe that a modest increase in the local concentration of endogenous agonists would result in the loss of effect of a highly potent competitive antagonist. It is more likely that the genetic deletion of an obligatory downstream mediator (CHOP) of the CB_1_ response is responsible for the loss of the effect of the antagonist. In this case the unchanged response to HFD may be mediated by a non-CB_1_ mechanism.

The reviewer is right. The elevated endocannabinoid 'tone' in the CHOP KO mice were not directly linked to the reduced effect of JD5037 in ameliorating dyslipidemia and hepatic steatosis. If our statement was misleading, we apologize for it. Based on our findings, CHOP plays an obligatory role in the CB_1_R-sOb-R cascade, and therefore, we agree with this reviewer that in CHOP KO animals the inability of JD5037 to ameliorate dyslipidemia is most likely related to the absence of CHOP. We amended the text accordingly (please see subsection “CHOP contributes to the metabolic response to peripheral CB_1_R blockade” and the Discussion).

3) A limitation of the impact of these findings is that they may not be extended to human physiology, as in humans circulating sOb-R is generated exclusively by ectodomain shedding catalyzed by the proteolytic enzymes ADAM10 and ADAM17, whose expression remained unchanged (Discussion). However, only mRNA was measured, which does not exclude possible changes in enzyme protein levels or enzyme activity. This limitation needs to be mentioned/discussed.

The regulation of ADAM10 and ADAM17 is complex and involves transcription, dynamic trafficking, cellular localization, and activity. Indeed, the current study focused on the transcriptional regulation of sOb-R by the CB_1_R/CHOP signaling pathway. However, other Ob-R isoforms, also expressed in humans, display (at the gene and protein levels) a trend similar to ObRe following either CB_1_R activation or blockade. These isoforms (Ob-Ra, Ob-Rb) are substrates for ectodomain shedding and therefore their transcriptional regulation may indirectly influence sObR levels and be relevant to human physiology. Nonetheless, the novel CB_1_R-induced regulation of CHOP levels and activity is not limited to animals and may also be relevant to humans. We have discussed these issues in the Discussion.

4) The authors demonstrate that obesity leads to loss of leptin sensitivity in the wild-type, but not in LCB1ko liver, as indicated by the loss of leptin-induced STAT3 phosphorylation in the former but not the latter. These important findings should be more prominently highlighted in the Discussion, which should also briefly touch on what's known about the effects of leptin in the liver (e.g. regulation of hepatic insulin sensitivity).

In addition to our findings in WT and LCB1 null mice, we assessed leptin sensitivity in animals with overexpression of CB_1_R only in hepatocytes (htgCB1^-/-^), and revealed an inability of leptin to increase STAT3 phosphorylation in these mice (please see the additional data in Figure 1R and S), further supporting the critical role of hepatic CB_1_R in regulating hepatic leptin sensitivity. As recommended by the reviewer, we have expanded the Introduction and Discussion accordingly.

Reviewer 2:1) There are points in the Introduction and Discussion that need however to be ameliorated, through minor but nevertheless important corrections, as detailed below:a) In general, the reader may wonder why the authors decided to investigate the role of CHOP in the effects of CB_1_R on sObR levels. This should be very briefly mentioned in the Introduction.

We expanded the Introduction to include information about the role of ER stress in the regulation of leptin resistance.

b) In the Introduction, the authors mention that the eCB system is overactive in obesity without quoting one of the first paper showing that this is true not only in the hypothalamus but also in peripheral cells, including the major source of leptin, i.e. the adipocytes (Matias et al., 2006). This should be mentioned; additionally, Monteleone et al., 2005, showed probably for the first time the elevation of circulating anandamide levels in overweight/obese women with binge eating disorder, again to be mentioned here;

The reviewer is right and these two citations as well as Matias et al., 2008 were added to the Introduction.

c) In the Discussion, when referring to leptin signaling in anorexia nervosa, the authors should again mention the Monteleone et al., 2005 study, showing how anandamide levels are also elevated in women with this disorder, possibly due to their low leptin levels;

We discussed this paper in the Discussion.

d) the above mentioned study by Matias et al., 2006, showed how both acute and chronic leptin significantly downregulates both anandamide and 2-AG levels in adipocytes (Figure 1 of that paper). This finding could be related to, or even explain, the authors' finding of higher eCB levels in CHOP-KO mice, which, by producing much less sObR, would counteract this inhibition of eCB levels also in adipocytes, by disinhibiting it. Since the authors did not provide a molecular mechanism for this effect, they may wish to discuss this possibility, especially if they have data showing that also in hepatocytes, like in adipocytes and hypothalamus, leptin decreases endocannabinoid levels via sObR (but also if they don't have such data).

Our data indeed show that CHOP null mice have a higher hepatic eCB 'tone', reduced levels of sOb-R, and a trend toward a reduction in circulating leptin levels (seen only in the STD-fed mice). These observations are in line with the reviewer's suggestion, and as mentioned in the text, deciphering the molecular mechanism by which CHOP regulates eCB production is out of the scope of this paper and will require the utilization of Ob-R-deficient models and/or the use of leptin antagonists to draw such a conclusion. Nevertheless, we amended the Discussion and included a possible explanation to these documented findings.

Reviewer 3:1) From a methodological viewpoint the study is well executed, however what I am missing a bit is the discussion of the physiologic significance and relevance of the findings for the field.

Thank you for finding our work to be well executed. We added a significance statement as well as discussed the relevance of our findings to human physiology in the Discussion (please also see our response to a similar comment raised by the first reviewer). In addition, the importance of our findings to the regulation of hepatic (peripheral) leptin resistance/sensitivity via CB_1_R was expanded on in the Discussion.

This is relevant, because the exact role of the soluble leptin receptor for leptin action is not fully resolved in the current literature. Since the liver-specific CB_1_R transgenic mice on CB_1_R-null background are resistant to diet induced obesity, despite lower soluble leptin receptor expression raises the question whether the observed phenomenon is relevant for overall leptin action, which is predominantly mediated via the CNS. In the CB_1_R transgenic mice central leptin action seems (based on the resistance to DIO) preserved or at least not majorly impaired, despite lower secretion of soluble leptin receptor, questioning the overall physiologic relevance of the observed reduction in soluble leptin receptor for whole body leptin action. Do the authors have additional data on leptin signaling (in liver and brain) in the transgenic mice?

Indeed, a soluble leptin receptor is only one of the molecular mechanisms that may contribute to leptin sensitivity/resistance. Here, we reported its role in regulating hepatic leptin resistance. The hTgCB1^-/-^ animals, despite producing less sOb-R, do not become obese because they lack CB_1_Rs everywhere except the liver. Central CB_1_Rs are required for the development of diet-induced obesity (DIO, as reported by Pang et al., Obesity, 2012), and are downstream of the effect that sOb-R has on leptin during the development of DIO, as expected. That is why hTgCB1^-/-^ animals, although having reduced sOb-R levels, are resistant to DIO. This is also true for mice globally lacking CB_1_Rs, which are resistant to DIO and its metabolic consequences, despite a similar caloric intake during the diet period (Ravinet Trillou et al., 2004; Osei-Hyiaman et al., 2005), which also points to energy metabolism being directly regulated by central endocannabinoids.

Nevertheless, this does not, by any means, imply that sOb-R is without relevance for DIO and energy metabolism in WT animals. On the contrary, the observations of Lou et al., 2010 strongly support the importance of sOb-R when CB_1_Rs are present. Here, we basically used hTgCB1^-/-^ animals to document the relevance of *hepatic* CB_1_R for the production of sOb-R, and show that indeed its overexpression in hepatocytes downregulates the expression of sOb-R, and that it contributes to hepatic leptin resistance. Following the reviewer's comment, we expanded our study and measured leptin signaling in the liver of hTgCB1^-/-^ animals, and found an inability of leptin to induce hepatic STAT3 phosphorylation in these mice. These additional results (please see the revised Figure 1R, S, the subsection “Hepatic CB_1_R regulates sOb-R levels and leptin signalling” and the Discussion) further imply a key role for CB_1_R in modulating hepatic leptin signaling via regulating sOb-R levels.

2) The claim that the regulation of soluble leptin receptor secretion via liver CB_1_R directly relates to liver leptin resistance is in my opinion not fully supported. The authors only present STAT3 phospho western blot data in CB_1_ KO and WT mice, but not the CB_1_R transgenic/CHOP KO mice. Direct effects of the endocannabinoid system on STAT3 signaling can thus not be excluded. I suggest deleting the claim from the title/Abstract. The authors should rather expand on this theory in the Discussion.

Thank you for this comment. As mentioned above, we have now provided evidence for the lack of hepatic leptin sensitivity in mice overexpressing CB_1_R in hepatocytes (see the revised Figure 1). We have also tested the ability of exogenous leptin to induce STAT3 phosphorylation in the liver of CHOP KO mice, and found reduced hepatic leptin sensitivity in the absence of CHOP (see the revised Figure 4G, H, and the subsection “CHOP plays a key role in the regulation of sOb-R by the eCB/CB_1_R system”), that is also in accordance with the reduced levels of sOb-R in these mice. We also discussed the relevance of hepatic leptin sensitivity and its regulation by CB_1_R (see the Discussion), and therefore decided to leave this claim in the title and Abstract. We hope that the reviewer, in light of the newly added data, also considers our claim now well-founded.

3) Circulating leptin levels (Figure 4F) in CHOP KO vs. WT JD5037 treated mice on HFD seem to correlate with fat mass (Figure 3F) rather than soluble leptin receptor levels. Are there any correlation data that would support the claim that soluble leptin receptor levels directly affects circulating leptin levels?

Many human and animal studies have demonstrated that sOb-R levels are inversely correlated with the plasma levels of leptin, BMI, and adiposity (Chan et al., 2002; Lahlou et al., 2000; Laimer et al., 2002; Ogier et al., 2002; Reinehr et al., 2005). The *positive* correlation between serum leptin concentrations and fat mass has been well demonstrated (Lonnqvist et al., 1995; Considine et al., 1996; Maffei et al., 1995). And indeed, the circulating leptin levels measured here in WT and CHOP KO mice (under STD, HFD, and HFD+JD5037) seem to be also *positively* correlated with the fat mass (see Author response image 2, left).

As pointed out by this reviewer, we found a *negative* correlation between hepatic sOb-R content and circulating leptin levels in both mouse strains (see Author response image 2, right), further supporting the idea that low levels of the hepatic soluble isoform contribute to obesity-related hyperleptinemia and subsequently, leptin resistance.
